# Exploring the Limits of Vision-Language-Action Manipulation in Cross-task Generalization

**Jiaming Zhou**
HKUST(GZ)

**Ke Ye**
HKUST(GZ)

**Jiayi Liu**
HKUST(GZ)

**Teli Ma**
HKUST(GZ)

**Zifan Wang**
HKUST(GZ)

**Ronghe Qiu**
HKUST(GZ)

**Kun-Yu Lin**
The University of Hong Kong

**Zhilin Zhao**
Sun Yat-sen University

**Junwei Liang**[*]
HKUST(GZ) and HKUST

## Abstract

The generalization capabilities of vision-language-action (VLA) models to unseen tasks are crucial to achieving general-purpose robotic manipulation in open-world settings. However, the cross-task generalization capabilities of existing VLA models remain significantly underexplored. To address this gap, we introduce **AGNOSTOS**, a novel simulation benchmark designed to rigorously evaluate zero-shot cross-task generalization in manipulation. AGNOSTOS comprises 23 unseen manipulation tasks for test, which are distinct from common training task distributions, and incorporates two levels of generalization difficulty to assess robustness. Our systematic evaluation reveals that current VLA models, despite being trained on diverse datasets, struggle to generalize effectively to these unseen tasks. To overcome this limitation, we propose **Cross-Task In-Context Manipulation (X-ICM)**, a method that conditions large language models (LLMs) on in-context demonstrations from seen tasks to predict action sequences for unseen tasks. Additionally, we introduce a **dynamics-guided sample selection** strategy that identifies relevant demonstrations by capturing cross-task dynamics. On AGNOSTOS, X-ICM significantly improves zero-shot cross-task generalization performance over leading VLA models, achieving improvements of 6.0% over $\pi_0$ [1] and 7.9% over VoxPoser [2]. We believe AGNOSTOS and X-ICM will serve as valuable tools for advancing general-purpose robotic manipulation. Project page: https://jiaming-zhou.github.io/AGNOSTOS/.

## 1   Introduction

Vision-Language-Action (VLA) models [3, 4, 5, 6, 7, 2, 8, 9, 1, 10] have motivated a new era of robotic manipulation by integrating visual perception, language understanding, and action generation. Through large-scale pre-training on diverse data, including human videos [11, 12], real or simulated cross-embodiment robotic demonstrations [3, 13, 14], VLA models can effectively generalize across visual variations *within known tasks* (i.e., within-task generalization of seen tasks), such as handling objects in novel scenes or with altered properties. However, the true promise of VLA models lies in their capacity to generalize *across* tasks: to handle previously unseen combinations of objects, goals, and actions without prior exposure. This capability, **zero-shot cross-task generalization**, is essential for real-world deployment, where robots are expected to tackle novel tasks as they arise dynamically.

Despite the rapid progress in VLA research, most prior work [3, 4, 8, 9, 1] has focused on generalization testing in real-world environments. However, these evaluations are typically non-reproducible, and rarely target cross-task (i.e., **unseen task**) zero-shot generalization. Recently, many

---

[*]Corresponding author: `junweiliang@hkust-gz.edu.cn`

39th Conference on Neural Information Processing Systems (NeurIPS 2025).

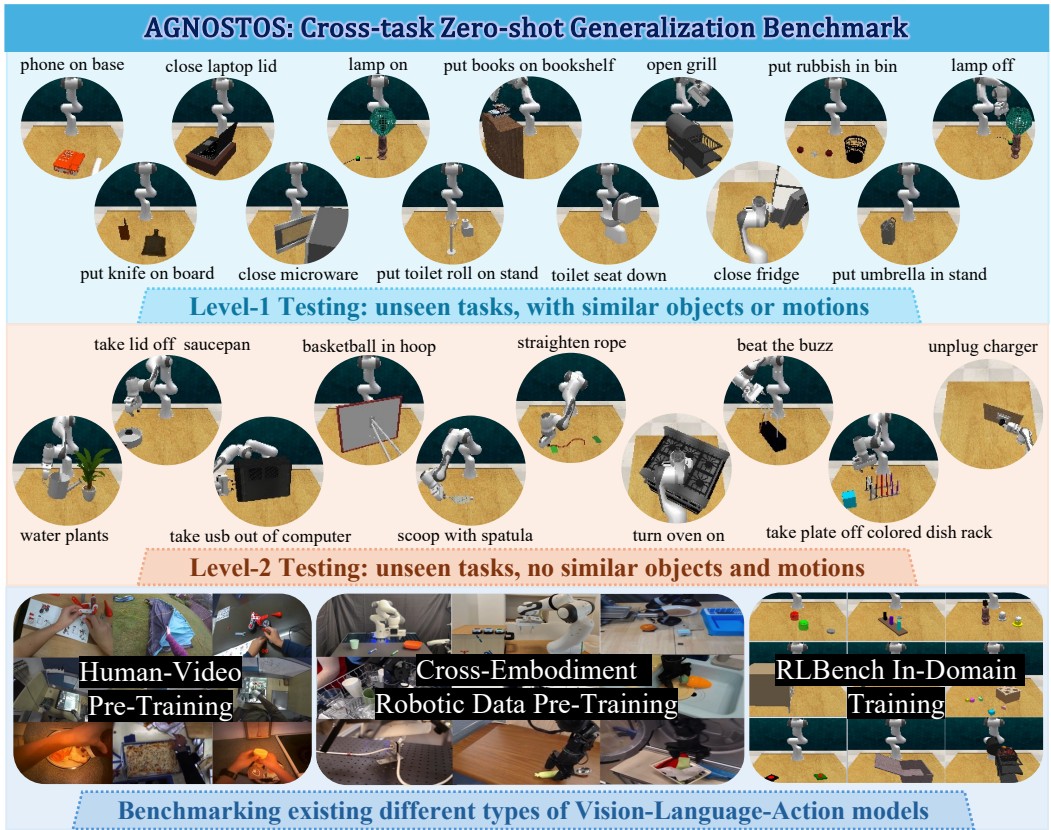

Figure 1: The proposed AGNOSTOS benchmark evaluates zero-shot cross-task generalization through two difficulty levels. Level-1 testing involves 13 unseen tasks sharing partial similarity (objects or motions) with seen tasks. Level-2 testing has 10 unseen tasks from entirely novel scenarios, requiring stronger generalization capabilities. We systematically assess three broad categories of vision-language-action models, revealing critical limits in their ability to adapt to unseen tasks.

works [15, 16, 17, 18, 19, 20, 21, 22, 23, 24, 25, 26] have been devoted to developing comprehensive simulated benchmarks, which could be utilized to evaluate the generalization of VLA models. While promising, these efforts mainly assess *within-task* generalization, leaving zero-shot cross-task generalization largely unexplored.

To address this critical gap, we present AGNOSTOS, a novel benchmark for evaluating zero-shot cross-task generalization in robotic manipulation. Built on RLBench [18], our benchmark comprises 23 **unseen** tasks that are carefully curated to differ from 18 commonly used **seen** training tasks [27, 28]. As shown in Figure 1, to probe different aspects of generalization, these unseen tasks are categorized into two difficulty levels:

- **Level-1** (13 tasks) shares partial semantics (e.g., similar objects like "cups" or motions like "put") with seen tasks.
- **Level-2** (10 tasks) introduces entirely novel scenarios with no overlapping objects or actions.

We benchmark three broad categories of VLA models:

1. Models [1, 2, 8, 9] trained on large-scale real-world robotic demonstrations [29] or built on multimodal large language models (MLLMs);
2. Models [30, 31, 32] pre-trained on large-scale human action video datasets [11];
3. Models [33, 27, 34] trained purely on in-domain RLBench data with advanced architectures.

Our empirical findings reveal a key limitation: **none** of the existing models generalize effectively to unseen tasks, highlighting the need for approaches that directly address cross-task generalization.

Motivated by the success of in-context learning [35, 36, 37] in large language models (LLMs), we propose a **Cross-task In-context Manipulation (X-ICM)** method to address the challenges of zero-shot cross-task generalization. X-ICM uses demonstrations from seen tasks as in-context examples to prompt LLMs to generate action plans for unseen tasks. A key challenge under this setup is selecting relevant demonstrations; irrelevant prompts fail to activate appropriate knowledge in the LLM, leading to poor cross-task predictions. To address this, we introduce a **dynamics-guided sample selection** strategy. This approach leverages learned representations of task dynamics—captured by predicting final observations from initial states and task descriptions—to identify relevant demonstrations across tasks. Guided by the learned dynamics, the dynamics-aware prompts can be formed that enable X-ICM to elicit high-quality action predictions from LLMs, even for unfamiliar, unseen tasks.

Our contributions are threefold:

- We introduce AGNOSTOS, the first benchmark to systematically evaluate zero-shot cross-task generalization in robotic manipulation for VLA models, with 23 unseen tasks spanning two levels of generalization difficulty.
- We propose X-ICM, a novel method that combines in-context prompting and dynamics-guided sample selection to enhance zero-shot cross-task transfer in VLA models.
- We conduct extensive evaluations of diverse VLA models on AGNOSTOS, uncovering fundamental limitations in current approaches and demonstrating the superior generalization of X-ICM.

## 2   Related Works

**Vision-Language-Action Models.** Generalizable robotic manipulation models [3, 4, 5, 38, 39, 40, 41, 42, 43, 44, 45, 46, 47, 48, 49, 50, 51], designed to enable robots to understand instructions and interact with the physical world, have largely followed two main paradigms. The first involves modular-based approaches [2, 52, 53, 54, 55, 56, 57, 58], where different MLLM components handle perception, language understanding, planning, and action execution. For example, VoxPoser [2] uses MLLMs to synthesize composable 3D value maps for manipulation. The second paradigm focuses on end-to-end approaches [3, 29, 9], training a policy model to directly map raw sensory inputs (e.g., vision, language instructions) to robot actions. The success of these models heavily depends on the scale and diversity of training data, which comes from various data sources, including large-scale human action videos [11] and large-scale cross-embodiment robotic data [29, 59]. Based on the data, OpenVLA [8] is the first fully open-sourced work that significantly promotes the development of the VLA community. $\pi_0$ [1] proposes a VLM-based flow matching architecture, which is trained on a diverse dataset from multiple dexterous robot platforms.

**Benchmarks for Vison-Language-Action Models.** Evaluating the generalization capabilities of VLA models requires robust and comprehensive benchmarks. While existing VLA models mainly focus on real-world testing, which suffers from reproducibility issues and rarely focuses systematically on zero-shot generalization to unseen tasks. Recently, many simulated benchmarks [15, 16, 17, 18, 19, 20, 21, 23, 24, 25, 26] have been proposed to offer a controlled environment for rigorous evaluation. Colosseum [24] enables evaluation of models across 14 axes of perturbations, including changes in color and texture, etc. GemBench [25] designs four levels of generalization, spanning novel placements, rigid and articulated objects, and complex long-horizon tasks. We find that these benchmarks have predominantly concentrated on evaluating visual variations **within tasks**, assessing how well models can generalize to new scenes or altered attributes of objects within the same task. However, these benchmarks do not focus on the more challenging aspect of **zero-shot cross-task evaluation**, where models must generalize to entirely new tasks with unseen combinations of object categories and motions. This gap in evaluation limits our understanding of the true generalization capabilities of VLA models. Motivated by this, this work proposes the first zero-shot cross-task manipulation benchmark, expanding the evaluation scope of the generalization capabilities of VLA models.

**In-context Learning with Large Language Models (LLMs).** LLMs [60, 35, 61, 62, 63, 64, 65, 66] have demonstrated a remarkable capability known as in-context learning (ICL) [35, 36, 37], where they can learn to perform new tasks based on a few examples provided within the prompt, without requiring updates to parameters. The potential of ICL is being explored in robotics [67, 68, 69, 70, 71, 72]. Prior works like KAT [67] and RoboPrompt [71] have shown that off-the-shelf LLMs can predict robot actions directly using within-task in-context samples. By extending their paradigms to a

cross-task setting, this work specifically focuses on the zero-shot cross-task generalization problem. Our X-ICM model uses demonstrations from seen tasks as in-context examples to prompt LLMs for action generation in completely unseen tasks. Under the zero-shot cross-task setting, the selection of in-context samples [73, 74, 75, 76] is crucial for a robust generalization. Our X-ICM method addresses this challenge by introducing a dynamics-guided sample selection strategy, ensuring that the selected seen demonstrations are relevant to the unseen tasks, thereby stimulating the cross-task generalization capabilities of LLMs.

## 3 AGNOSTOS: Zero-shot Cross-task Generalization Benchmark

To systematically assess the **zero-shot cross-task generalization** ability of vision-language-action (VLA) models, we introduce AGNOSTOS, a reproducible benchmark built upon the RLBench simulation environment. AGNOSTOS features 18 seen tasks for training and 23 unseen tasks for rigorous generalization testing.

**Training**. For training, we adopt the standard set of 18 RLBench tasks that are widely used in prior work [27, 28]. Examples of these seen tasks are shown in Figure A1. We collect 200 language-conditioned demonstrations per task, resulting in 3600 demonstrations in total. These demonstrations enable VLA models to be fine-tuned to reduce the domain and embodiment gaps between pre-training data and RLBench data.

**Testing.** As illustrated in Figure 1, AGNOSTOS comprises 23 held-out unseen tasks with semantics that are disjoint from the seen set (videos of all tasks are available in the Supplementary Materials). We categorize the unseen tasks into two difficulty levels. **Level-1**: 13 tasks that share partial semantic similarity with seen tasks—either in object types (e.g., cups) or motion primitives (e.g., stacking). **Level-2**: 10 tasks that exhibit no overlap in either object categories or motion types, requiring broader compositional reasoning and semantic extrapolation. Details on task curation and difficulty categorization are provided in Section A1.1 of the Appendix. For each unseen task, we perform three test runs with different seeds, each consisting of 25 rollouts, and report the mean and standard deviation of success rates.

To explore generalization boundaries, AGNOSTOS evaluates three broad families of VLA[2] models:

1. **Foundation VLA models**: trained on large-scale real-world cross-embodiment robotic data [29] or built upon LLM or VLM models, including OpenVLA [8], RDT [9], $\pi_0$ [1], LLARVA [77], SAM2Act [34], 3D-LOTUS++ [25], and VoxPoser [2].

2. **Human-video VLA models**: pre-trained on large-scale human action videos [11] to capture rich human-object interactions for downstream robotic fine-tuning, including R3M [30], D4R [31], R3M-Align [32], and D4R-Align [32].

3. **In-domain VLA models**: trained from scratch on RLBench's 18 seen tasks with task-specific model architectures. These serve as strong baselines without domain mismatch, including PerAct [27], RVT [28], RVT2 [33], Sigma-Agent [78], and Instant Policy [69].

To ensure a fair comparison, we fine-tune all models on the same 18 seen tasks when the models involve the embodiment gaps, following the official protocols of each method. A detailed description of the fine-tuning process and evaluation on AGNOSTOS is provided in Section A1.2 of the Appendix. Table 1 compares AGNOSTOS with existing robotic manipulation benchmarks in terms of their support for cross-task evaluation, i.e., considering test tasks that have different object categories or motions from training tasks. While many benchmarks focus on within-task visual generalization, few explicitly support **zero-shot cross-task generalization test**, and even fewer include tasks as challenging as our Level-2 testing scenarios. Even benchmarks that include some form of cross-task evaluation[3] often test on a narrow set of tasks, and typically evaluate only in-domain models, neglecting foundation and human-video pre-trained VLA models. AGNOSTOS fills this gap by **supporting broad model coverage and introducing novel, difficult tasks in Level-2**, providing a more comprehensive and diagnostic assessment of cross-task generalization.

---

[2]We only consider evaluating the VLA models that are fully open-source.

[3]We carefully categorize their test tasks into our defined two-level difficulty task sets.

Table 1: Comparison of manipulation benchmarks that focuses on cross-task generalization testing. The comparison evaluates whether each benchmark includes cross-task testing, the types of VLA models evaluated, and the number of Level-1 and Level-2 unseen tasks included.

| Benchmark | Simulator | No. of train tasks | No. of test tasks | Cross-task Zero-shot Generalization Test | | | | | |
|---|---|---|---|---|---|---|---|---|---|
| | | | | Cross task | Evaluated VLA models | | | Level-1 unseen tasks | Level-2 unseen tasks |
| | | | | | In-domain | Human-video | Foundation | | |
| RLBench-18Task [27] | RLBench | 18 | 18 | ✗ | - | - | - | - | - |
| CALVIN [17] | PyBullet [79] | 34 | 34 | ✗ | - | - | - | - | - |
| Colosseum [24] | RLBench | 20 | 20 | ✗ | - | - | - | - | - |
| VLMBench [23] | RLBench | 8 | 8 | ✗ | - | - | - | - | - |
| Ravens [80] | PyBullet [79] | 10 | 10 | ✓ | ✓ | ✗ | ✗ | 3 | 0 |
| VIMA-Bench [81] | Ravens | 13 | 17 | ✓ | ✓ | ✗ | ✗ | 4 | 0 |
| GemBench [25] | RLBench | 16 | 44 | ✓ | ✓ | ✗ | ✗ | 15 | 0 |
| **AGNOSTOS (Ours)** | RLBench | 18 | 23 | ✓ | ✓ | ✓ | ✓ | 13 | 10 |

# 4 X-ICM: Cross-task In-context Manipulation Method

To push the boundaries of zero-shot cross-task generalization in vision-language-action (VLA) models, we propose a method called Cross-task In-context Manipulation (X-ICM). Leveraging the cross-task generalization capabilities of LLMs, X-ICM utilizes demonstrations from seen tasks as in-context examples. The dynamic characteristics of these examples are used to prompt the LLM to predict action sequences for unseen tasks. In contrast to prior works [71, 67] that apply LLMs' in-context learning to within-task generalization, our work is the first to extend this paradigm to a **zero-shot cross-task setting**. A central challenge in this setting is that the selection of in-context demonstrations significantly affects generalization performance. To address this, we design a **dynamics-guided sample selection** module that measures similarities between dynamic representations of seen and unseen tasks to guide the selection process, resulting in improved cross-task generalization.

## 4.1 Problem Definition and Method Overview

We tackle the problem of zero-shot cross-task robotic manipulation by exploiting the in-context learning ability of off-the-shelf LLMs. We assume access to a dataset of demonstrations from seen tasks, denoted as $\mathcal{D}^s = \{V_i^s, A_i^s, L_i^s\}_{i=1}^N$, where $N$ is the total number of seen demonstrations, $V_i^s = \{v_{i,1}^s, \cdots, v_{i,T}^s\}$, $A_i^s = \{a_{i,1}^s, \cdots, a_{i,T}^s\}$, and $L_i^s$ are the visual observation sequence, action sequence, and language description of the $i$-th seen demonstration ($T$ is the sequence length, which varies for each demonstration). For an unseen task with the given initial visual observation $v^u$, and language description $L^u$, our zero-shot cross-task manipulation method can be formulated as follows:

$$A_{pred}^u = \{a_1^u, \cdots, a_t^u\} = LLM(\mathcal{P}(\mathcal{F}^{sel}(\mathcal{D}^s), v^u, L^u)), \tag{1}$$

where $\mathcal{F}^{sel}$ is the cross-task in-context demonstration selection process, $\mathcal{P}$ is the cross-task in-context prompt construction process, and $A_{pred}^u$ is the predicted action sequence for the tested unseen task.

Figure 2 outlines our X-ICM framework, which comprises two core modules. The **Dynamics-guided Sample Selection module** introduces a dynamics diffusion model, and then retrieves effective demonstrations based on dynamic similarities between seen and unseen demonstrations. The **Cross-task In-context Prediction module** constructs the LLM prompt using retrieved demonstrations to predict action sequences for the unseen task. Next, we describe each module in detail below.

## 4.2 Dynamics-guided Sample Selection module

Selecting effective in-context examples is essential for achieving robust cross-task generalization. We propose a two-stage Dynamics-guided Sample Selection module that learns dynamic representations from demonstrations using diffusion models, and retrieves examples that are most relevant to the current unseen task based on the similarities between the dynamic representations of demonstrations.

**Diffusion-based dynamics modeling stage.** To effectively capture the dynamic representations within each demonstration, we train a dynamics diffusion model $\mathcal{G}$ on all seen demonstrations. For each demonstration (e.g., the $i$-th), the dynamics diffusion model $\mathcal{G}$ takes the initial visual observation

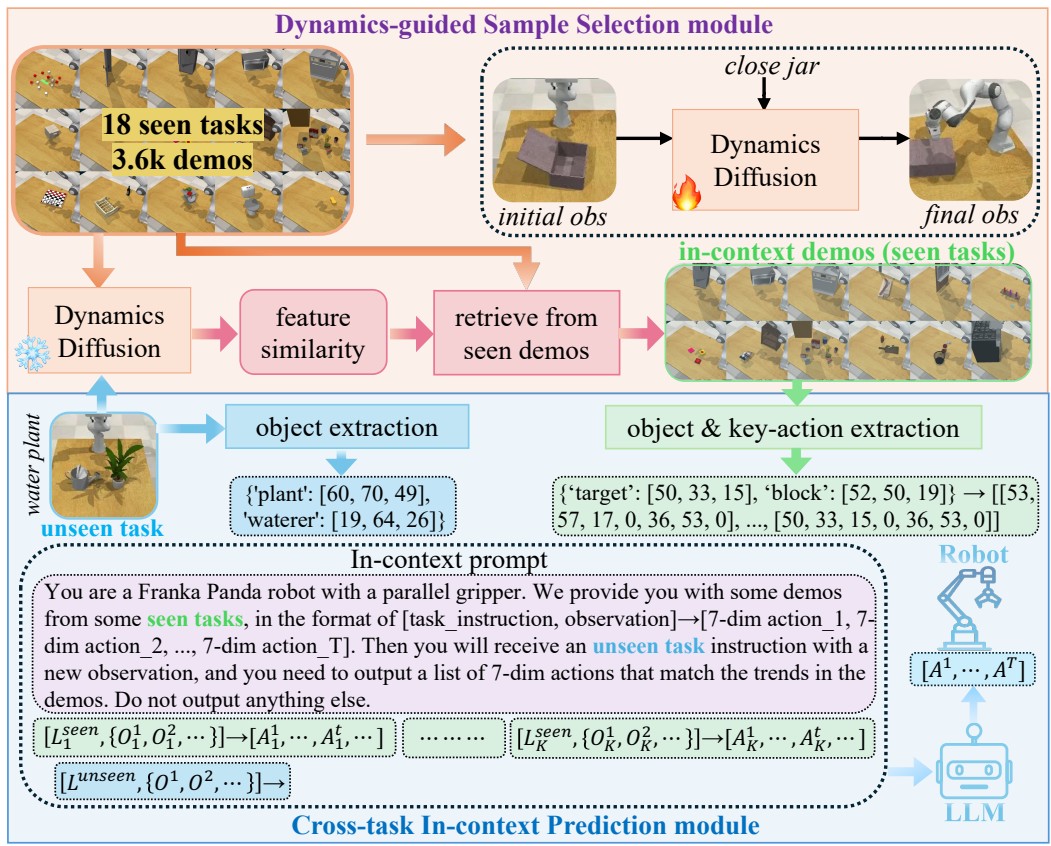

Figure 2: **X-ICM Method Overview.** X-ICM employs a dynamics-guided sample selection module to retrieve effective demonstrations from seen tasks for each tested unseen task. These demonstrations are then used by the cross-task in-context prediction module to construct the prompt that drives the LLM to predict the corresponding action sequence.

$v_{i,1}^s$ and language description $L_i^s$ as inputs, and predicts the future observation that matches the final visual observation $v_{i,T}^s$. The model $\mathcal{G}$ is initialized from InstructPix2Pix [82] and is optimized by:

$$\min_{\mathcal{G}} \mathbb{E}_{i,z,\epsilon\sim\mathcal{N}(0,I)} \left[ \left\| \epsilon - \epsilon_{\mathcal{G}}(v_{i,T,z}^s, z, v_{i,1}^s, L_i^s) \right\|_2^2 \right], \tag{2}$$

where $z$ is the diffusion timestep, $\epsilon$ is the noise, $v_{i,T,z}^s$ is the noised final observation, and $\epsilon_{\mathcal{G}}$ is the noise predictor in model $\mathcal{G}$. By predicting the future, the inherent dynamics of each demonstration can be effectively modeled. Figure A6 shows the generation results of our dynamics diffusion model.

**Dynamics-guided retrieving stage.** Once trained, we use $\mathcal{G}$ to extract a dynamic feature $f_i^s$ for the $i$-th seen demonstration:

$$f_i^s = [f_i^{s,vis}, f_i^{s,lang}] = \mathcal{G}(v_{i,1}^s, L_i^s) \in \mathbb{R}^{2\times1024}, \tag{3}$$

where $f_i^{s,vis} \in \mathbb{R}^{1024}$ is the predicted latent feature of the target visual observation, $f_i^{s,lang} \in \mathbb{R}^{1024}$ is the textual feature of the language description. For the unseen task, we similarly compute its feature $f^u$. Then, we compute cosine similarities between features and select the top-$K$ seen demonstrations with the highest similarity:

$$R_i = \frac{f^u \cdot f_i^s}{\|f^u\|\|f_i^s\|} \quad i \in \{1, \dots, N\}, \tag{4}$$

$$\mathcal{I}_{idx} = \arg \operatorname{topk}_{i\in\{1,\dots,N\}}^K (R_i), \tag{5}$$

where $R_i$ represents the cross-task dynamics similarity score between the tested unseen task and the $i$-th seen demonstration. And the set $\mathcal{I}_{idx}$ contains the indices of $K$ seen demonstrations that achieve the highest similarity scores. These $K$ seen demonstrations are retrieved to construct the cross-task in-context prompt.

### 4.3 Cross-task In-context Prediction module

For a tested unseen task, we use the obtained $K$ most-relevant seen demonstrations to construct the in-context prompt for cross-task action prediction with LLMs. Our Cross-task In-context Prediction module follows the practices in existing within-task in-context manipulation methods [71, 67], but we are the first to extend this paradigm to the zero-shot cross-task setting.

**Textualizing key information.** As shown at the bottom of Figure 2, for each selected seen demonstration, we extract the 3D positions of the centers of objects and the key-action sequence. For the 3D positions of objects' centers, we use GroundingDINO [83] to detect the 2D positions of objects, and then use the depth, intrinsic, and extrinsic information of the camera to obtain the 3D coordinates of objects' centers under the robot-base coordinate system. Additionally, we evenly divide the workspace into $100 \times 100 \times 100$ grids, and then normalize the 3D coordinates of the objects' centers into the grid's coordinate system. Thus, for the $m$-th object $O_k^m$ in the $k$-th seen demonstration, we textualize it as $O_k^m = $ "$objectname$: $[x, y, z]$" (e.g., "block: [52, 50, 19]"). For the key-action sequence, following [84], we select the actions when the gripper state changes or the joint velocities are near zero. In this way, the execution process of a demonstration can be determined by several key-actions. We use the 7-dimensional state of the end-effector to represent the robot action, i.e., the 3D positions, the roll-pitch-yaw angles, and the gripper's open/close. The 3D positions of the end-effector are also normalized into the grid's coordinate system of the workspace. And the angles of end-effector are discretized into 72 bins, with each occupying 5 degrees. Thus, for the $t$-th key-action $A_k^t$ in the $k$-th seen demonstration, we textualize it as $A_k^t = $ "$[x, y, z, roll, pitch, yaw, gripper]$" (e.g., "[53, 57, 17, 0, 36, 53, 0]").

**Constructing in-context prompt.** With the above textualization of object and key-action information, for each seen demonstration (e.g., the $k$-th), we can textualize it as the mapping from its language description and object context to its key-action context:

$$\text{``}[L_k^{seen}, \{O_k^1, \cdots, O_k^m, \cdots\}] \rightarrow [A_k^1, \cdots, A_k^t, \cdots]\text{''}. \tag{6}$$

We textualize all $K$ selected seen demonstrations in this manner, and concatenate them in descending order based on their dynamics similarity scores to the tested unseen task. For the tested unseen task, we textualize it using its language description and object context, i.e., "$[L^{unseen}, \{O^1, \cdots, O^m, \cdots\}] \rightarrow$". Finally, the cross-task in-context prompt is formed by concatenating the system prompt, the textualized all seen demonstrations, and the textualized tested unseen task (see the bottom of Figure 2 and an example in Figure A7 of the Appendix). This prompt will serve as the input of LLMs, which incentivizes the cross-task generalization capability of LLMs to predict the key-action sequence for the tested unseen task.

## 5 Experiments

**Implementation Details.** For our X-ICM method on the AGNOSTOS benchmark, we use a total of $N = 3600$ seen demonstrations. During in-context prompt construction, we select $K = 18$ demonstrations. We mainly use off-the-shelf Qwen2.5-Instruct [65] models with 7B and 72B parameters, referred to as X-ICM (7B) and X-ICM (72B), respectively. These are deployed using two or eight A6000 GPUs. For a fair comparison with existing zero-shot baselines (e.g., VoxPoser [2]), in simulation we use the ground-truth positions of objects. Ablation on using different sizes of LLMs is presented in Sec A2.2 of the Appendix.

### 5.1 Benchmarking Vision-Language-Action Models

Table 2 presents the zero-shot cross-task performance of our X-ICM models on 23 unseen tasks from the benchmark, including 13 Level-1 and 10 Level-2 tasks. These tasks are designed to evaluate the generalization ability of VLA models under varying levels of difficulty. We compare our models against a diverse set of baselines, including models trained on in-domain RLBench data, human videos, and LLM/VLM-based foundations. Key observations include:

- X-ICM (7B) and X-ICM (72B) achieve average success rates of 23.5% and 30.1%, respectively, outperforming all existing VLA models.
- On Level-1 tasks, X-ICM (7B) surpasses the prior SoTA $\pi_0$ [1] by 6.9%. For Level-2 tasks, performance gains are more pronounced with the 72B model.

Table 2: Cross-task zero-shot manipulation performance on 23 unseen tasks, where the column headers show the abbreviation of each unseen task (see full task names in Table A1). N/A indicates that the tested tasks are removed since they overlap with the training tasks of the methods. Level-1 and Level-2 represent tasks with two different difficulty levels. The prefix * indicates second best.

| | methods | Level-1 tasks | | | | | | | | | | | | |
|---|---|---|---|---|---|---|---|---|---|---|---|---|---|---|
| | | Toilet | Knife | Fridge | Microwave | Laptop | Phone | Seat | LampOff | LampOn | Book | Umbrella | Grill | Bin |
| in-domain training | PerAct [27] | 0.0 | 5.3 | 37.3 | 64.0 | 2.7 | 0.0 | 72.0 | 0.0 | 1.3 | 0.0 | 1.3 | 8.0 | 54.7 |
| | RVT [28] | 0.0 | 2.7 | 50.7 | 26.7 | 50.7 | 2.7 | 40.0 | 0.0 | 1.3 | 0.0 | 1.3 | 0.0 | 6.7 |
| | Sigma-Agent [78] | 0.0 | 9.3 | 56.0 | 9.3 | 30.7 | 1.3 | 65.3 | 1.3 | 0.0 | 0.0 | 0.0 | 1.3 | 4.0 |
| | RVT2 [33] | 0.0 | 1.3 | 0.0 | 17.3 | 42.7 | 1.3 | 62.7 | 2.7 | 1.3 | 0.0 | 1.3 | 5.3 | 34.7 |
| | InstantPolicy [69] | 0.0 | 1.3 | 13.3 | 4.0 | 4.0 | 18.7 | 24.0 | 0.0 | 0.0 | 0.0 | 0.0 | 0.0 | 0.0 |
| human-video pretraining | D4R [31] | 0.0 | 8.0 | 32.0 | 30.7 | 24.0 | 0.0 | 65.3 | 20.0 | 4.0 | 0.0 | 0.0 | 0.0 | 0.0 |
| | R3M [30] | 0.0 | 0.0 | 37.3 | 22.7 | 25.3 | 1.3 | 62.7 | 6.7 | 4.0 | 0.0 | 0.0 | 0.0 | 0.0 |
| | D4R-Align [32] | 0.0 | 2.7 | 45.3 | 74.7 | 24.0 | 0.0 | 41.3 | 0.0 | 0.0 | 1.3 | 0.0 | 0.0 | 0.0 |
| | R3M-Align [32] | 0.0 | 4.0 | 49.3 | 25.3 | 21.3 | 0.0 | 49.3 | 0.0 | 5.3 | 0.0 | 0.0 | 1.3 | 1.3 |
| VLA foundations | OpenVLA [8] | 0.0 | 5.3 | 38.7 | 40.0 | 57.3 | 0.0 | 53.3 | 12.0 | 1.3 | 1.3 | 0.0 | 10.7 | 0.0 |
| | RDT [9] | 0.0 | 0.0 | 46.7 | 13.3 | 14.7 | 0.0 | 50.7 | 0.0 | 0.0 | 1.3 | 0.0 | 8.0 | 0.0 |
| | $\pi_0$ [1] | 0.0 | 5.3 | 85.3 | 24.0 | 40.0 | 1.3 | 64.0 | 18.7 | 8.0 | 1.3 | 0.0 | 33.3 | 1.3 |
| | LLARVA [77] | 0.0 | 0.0 | 12.0 | 0.0 | 6.7 | 0.0 | 40.0 | 0.0 | 0.0 | 0.0 | 0.0 | 0.0 | 0.0 |
| | 3D-LOTUS [25] | 0.0 | 6.7 | N/A | N/A | N/A | 0.0 | 6.7 | 0.0 | 0.0 | 0.0 | 0.0 | 13.3 | 5.3 |
| | 3D-LOTUS++ [25] | 0.0 | 5.3 | N/A | N/A | N/A | 9.3 | 68.0 | 10.7 | 0.0 | 0.0 | 0.0 | 29.3 | 13.3 |
| | SAM2Act [34] | 0.0 | 0.0 | 36.0 | 40.0 | 6.7 | 6.7 | 62.7 | 6.7 | 0.0 | 1.3 | 1.3 | 9.3 | 0.0 |
| | VoxPoser [2] | 0.0 | 0.0 | 0.0 | 0.0 | 5.3 | 8.0 | 28.0 | 88.7 | 25.3 | 0.0 | 0.0 | 0.0 | 82.7 |
| Ours | **X-ICM (7B)** | 1.3 | 26.7 | 22.7 | 45.3 | 33.3 | 57.3 | 48.0 | 58.7 | 50.7 | 1.3 | 0.0 | 8.0 | 18.7 |
| | **X-ICM (72B)** | 6.7 | 69.3 | 12.7 | 58.7 | 34.0 | 68.0 | 51.3 | 86.7 | 74.7 | 2.0 | 1.3 | 5.3 | 18.7 |

| | methods | Level-2 tasks | | | | | | | | | | Level-1 avg (std) | Level-2 avg (std) | All avg (std) |
|---|---|---|---|---|---|---|---|---|---|---|---|---|---|---|
| | | USB | Lid | Plate | Ball | Scoop | Rope | Oven | Buzz | Plants | Charger | | | |
| in-domain training | PerAct [27] | 58.7 | 2.7 | 0.0 | 0.0 | 0.0 | 0.0 | 1.3 | 4.0 | 6.7 | 2.7 | 19.0 (1.4) | 7.6 (1.1) | 14.0 (0.9) |
| | RVT [28] | 89.3 | 2.7 | 0.0 | 0.0 | 0.0 | 0.0 | 4.0 | 8.0 | 5.3 | 4.0 | 14.0 (1.4) | 11.3 (1.6) | 12.8 (0.2) |
| | Sigma-Agent [78] | 88.0 | 0.0 | 0.0 | 0.0 | 0.0 | 0.0 | 4.0 | 8.0 | 5.3 | 1.3 | 13.7 (1.6) | 10.7 (1.7) | 12.4 (0.4) |
| | RVT2 [33] | 22.7 | 40.0 | 0.0 | 0.0 | 0.0 | 0.0 | 0.0 | 1.3 | 1.3 | 1.3 | 13.1 (0.4) | 6.7 (1.3) | 10.3 (0.6) |
| | InstantPolicy [69] | 26.7 | 1.3 | 0.0 | 0.0 | 0.0 | 0.0 | 0.0 | 1.3 | 0.0 | 0.0 | 4.3 (4.2) | 2.9 (1.4) | 3.7 (3.0) |
| human-video pretraining | D4R [31] | 98.7 | 0.0 | 0.0 | 0.0 | 0.0 | 0.0 | 1.3 | 1.3 | 1.3 | 4.0 | 14.1 (0.3) | 10.7 (0.2) | 12.6 (0.2) |
| | R3M [30] | 48.0 | 0.0 | 0.0 | 0.0 | 0.0 | 0.0 | 8.0 | 2.7 | 2.7 | 1.3 | 12.3 (1.4) | 6.3 (0.9) | 9.7 (0.6) |
| | D4R-Align [32] | 89.3 | 1.3 | 0.0 | 0.0 | 0.0 | 0.0 | 8.0 | 6.7 | 0.0 | 1.3 | 14.5 (1.0) | 10.7 (0.2) | 12.8 (0.6) |
| | R3M-Align [32] | 90.7 | 0.0 | 1.3 | 0.0 | 0.0 | 0.0 | 2.7 | 13.3 | 4.0 | 0.0 | 12.9 (0.7) | 11.2 (0.7) | 12.2 (0.3) |
| VLA foundations | OpenVLA [8] | 77.3 | 0.0 | 0.0 | 0.0 | 0.0 | 0.0 | 6.7 | 5.3 | 2.7 | 0.0 | 16.9 (1.3) | 9.2 (0.7) | 13.6 (0.8) |
| | RDT [9] | 100.0 | 29.3 | 4.0 | 0.0 | 0.0 | 0.0 | 8.0 | 2.7 | 0.0 | 0.0 | 10.4 (0.5) | 14.4 (0.9) | 12.1 (0.4) |
| | $\pi_0$ [1] | 97.3 | 0.0 | 1.3 | 0.0 | 0.0 | 0.0 | 14.7 | 5.3 | 1.3 | 0.0 | *21.7 (0.4) | 12.0 (0.9) | *17.5 (0.4) |
| | LLARVA [77] | 24.0 | 0.0 | 0.0 | 0.0 | 0.0 | 0.0 | 0.0 | 0.0 | 0.0 | 0.0 | 4.5 (0.1) | 2.4 (0.0) | 3.6 (0.1) |
| | 3D-LOTUS [25] | 85.3 | 0.0 | 1.3 | 0.0 | 0.0 | 0.0 | 0.0 | 4.0 | 0.0 | 0.0 | 3.2 (0.5) | 9.1 (0.7) | 6.2 (0.5) |
| | 3D-LOTUS++ [25] | 90.7 | 30.7 | 0.0 | 0.0 | 5.3 | 1.3 | 8.0 | 8.7 | 6.7 | 0.0 | 13.6 (1.0) | 15.1 (1.1) | 14.4 (1.0) |
| | SAM2Act [34] | 92.0 | 49.3 | 0.0 | 0.0 | 0.0 | 0.0 | 1.3 | 6.7 | 4.0 | 5.3 | 13.1 (0.4) | *15.9 (1.3) | 14.0 (0.7) |
| | VoxPoser [2] | 32.0 | 76.0 | 0.0 | 8.0 | 0.0 | 0.0 | 0.0 | 1.3 | 0.0 | 4.0 | 18.1 (0.4) | 12.1 (0.4) | 15.6 (0.2) |
| Ours | **X-ICM (7B)** | 98.7 | 20.0 | 6.7 | 9.3 | 0.0 | 6.7 | 16.0 | 2.7 | 5.3 | 4.0 | **28.6 (1.9)** | **16.9 (1.3)** | **23.5 (1.6)** |
| | **X-ICM (72B)** | 98.7 | 13.3 | 4.7 | 36.0 | 0.7 | 16.0 | 20.7 | 7.3 | 2.7 | 2.7 | **37.6 (1.4)** | **20.3 (1.7)** | **30.1 (1.0)** |

- While some VLA foundation models show decent performance, particularly $\pi_0$ on Level-1 tasks and SAM2Act [34]/3D-LOTUS++ [25] on Level-2 tasks, they fall short of the broad generalization capabilities demonstrated by our X-ICM models.

- All prior models completely fail on at least eight of the 23 tasks. In contrast, X-ICM (7B) fails on only two, and X-ICM (72B) succeeds on all.

## 5.2 Ablation Studies

**Dynamics-guided sample selection.** We assess the impact of the dynamics-guided sample selection module by comparing X-ICM (72B) with and without it (denoted as *w/o sel*, using random sampling) at the top-left of Table 3. Incorporating the module notably boosts performance and robustness across both task levels. This confirms its effectiveness in identifying informative demonstrations that enhance the LLM's cross-task generalization ability. Further analysis, including visualizations from the dynamics diffusion model and comparisons of dynamic features, are provided in Appendix A2.1.

Table 3: Effects of dynamics-guided sample selection module and different model sizes.

| Models | Level-1 | Level-2 | All |
|---|---|---|---|
| X-ICM (72B) *w/o sel* | 30.7 (4.7) | 18.0 (2.2) | 25.2 (3.2) |
| X-ICM (72B) | 37.6 (1.4) | 20.3 (1.7) | 30.1 (1.0) |

Table 4: Effect of using different LLM models.

| LLMs | Level-1 | Level-2 | All |
|---|---|---|---|
| Deepseek-R1-Distill-Qwen-7B | 10.7 (1.1) | 7.9 (0.5) | 9.5 (0.5) |
| Llama3.0-8B-Instruct | 17.4 (0.7) | 11.7 (1.6) | 15.1 (0.3) |
| Ministral-8B-Instruct-2410 | 22.9 (0.7) | 14.8 (0.3) | 19.5 (0.5) |
| InternLM3-8B-Instruct | 27.9 (0.7) | 13.3 (0.4) | 21.8 (0.5) |
| Qwen2.5-7B-Instruct | **28.6 (1.9)** | **16.9 (1.3)** | **23.5 (1.6)** |

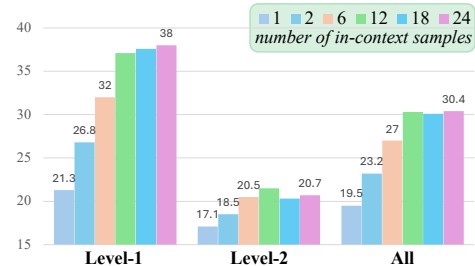

Figure 3: The effect of varying the number of in-context demonstrations.

**Effect of the number of in-context demonstrations.** We explore how performance varies with the number of in-context demonstrations using the Qwen2.5-72B-Instruct model. Figure 3 shows that performance improves rapidly as the number increases from 1 to 12, after which it plateaus. This suggests that a moderate number of relevant demonstrations is critical. Too many may introduce irrelevant noise and diminish gains.

**Comparison of LLM backbones.** Table 4 compares the performance of X-ICM (7B) when replacing Qwen2.5-7B-Instruct with alternative 7B/8B models, including Deepseek-R1-Distill-Qwen-7B [85], Llama3.0-8B-Instruct [86], Ministral-8B-Instruct-2410 [87], and InternLM3-8B-Instruct [88]. The results show significant variability in performance, underscoring the importance of choosing a powerful and well-aligned LLM backbone.

## 5.3 Real-world Experiments

To evaluate real-world applicability, we test X-ICM on five physical manipulation tasks using an xArm7 robot arm, DH-Robotics gripper, and a third-person Orbbec camera: *put block into bin*, *push button*, *put bottle into box*, *stack blocks*, and *stack cups*. We collect 20 demonstrations for each of the five tasks. During testing, we use the demonstrations from the other four tasks to construct the cross-task in-context prompt, enabling zero-shot generalization. The Qwen2.5-72B-Instruct LLM is used as the backbone and the number of seen demonstrations is set to 18. Each task is executed 20 times, and we report average success rates. Results shown at the bottom of Figure 4 demonstrate the strong real-world **zero-shot cross-task** performance of our approach. We provide additional details in Section A3 of the Appendix, and showcase some testing videos that include both successful and failed cases in the Supplementary Materials.

In addition, we also evaluate the proposed X-ICM model on long-horizon tasks under the zero-shot cross-task setting. We use tasks *put block into bin*, *push button*, *put bottle into box*, *stack blocks*, and *pull out the middle block* as seen tasks, where we have 10 demonstrations for each task. The long-horizon task we evaluated is *clean the table*, according the robot's visual observation, we decompose the long-horizon task into three sequential sub-tasks, ie, *stack cups*, *put the stacked cups into plate*, *place the mango on top of the stacked cups*. We evaluated this task over 20 rollouts, with sub-task success contingent on the success of all prior sub-tasks. For our X-ICM method, the success rates on sub-tasks "stack cups", "put the stacked cups into plate", "place the mango on top of the stacked cups." are 40%, 25%, and 5%, respectively. Thus, the overall success rate for the long-horizon task is 5%. These results align with expectations, as the sequential nature of long-horizon tasks leads to cumulative failure. The cross-task zero-shot setting exacerbates these difficulties. These findings highlight the need for more advanced algorithms to handle long-horizon tasks in zero-shot scenarios.

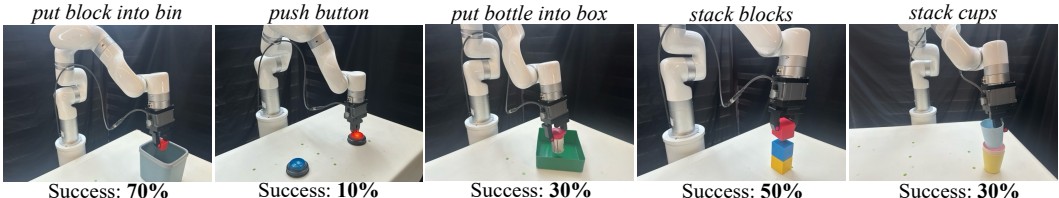

| *put block into bin* | *push button* | *put bottle into box* | *stack blocks* | *stack cups* |
| Success: **70%** | Success: **10%** | Success: **30%** | Success: **50%** | Success: **30%** |

Figure 4: **Results of five real-world tasks.** The tests are conducted in a zero-shot cross-task manner.

# 6 Conclusions and Discussions

**Conclusions.** In this work, we have introduced AGNOSTOS, a benchmark for evaluating zero-shot cross-task generalization in robotic manipulation. With 23 unseen tasks across two difficulty levels, AGNOSTOS provides a rigorous testbed for assessing the limits of Vision-Language-Action (VLA) models. Our evaluation of diverse VLA models reveals their significant limitations in unseen task generalization. To address this, we propose X-ICM, a cross-task in-context manipulation method that leverages the cross-task generalization capabilities of LLMs. The X-ICM achieves substantial improvements over existing VLA models, demonstrating robust zero-shot cross-task generalization.

**Discussions.** While X-ICM significantly enhances zero-shot cross-task generalization, its performance on many unseen tasks, particularly those with both novel objects and motion primitives, remains limited due to LLMs' challenges in extrapolating beyond pre-training data and in-context demonstrations. In addition, the use of visual information is limited to textualizing object information, which may ignore important visual context in the raw data. To address these limitations, future work could integrate multi-modal reasoning, combining vision, language, and action data to improve generalization to novel semantics. Additionally, leveraging generalizable concepts like object trajectories as a bridge between MLLMs' perception and dynamic action sequence prediction could reduce cross-task generalization complexity. Scaling X-ICM to diverse robotic embodiments would further enhance its versatility across varied platforms and sensor configurations. We believe the proposed X-ICM benchmark and X-ICM method will inspire future research in generalizable robotic manipulation, paving the way for robots that can seamlessly adapt to open-world environments.

**Acknowledgements.** This work was supported by the National Natural Science Foundation of China (No. 62306257), the Guangzhou Municipal Science and Technology Project (No. 2024A03J0619 and No. 2024A04J4390) and the Guangzhou-HKUST(GZ) Joint Funding Program (Grant No.2023A03J0008), Education Bureau of Guangzhou Municipality.

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

# Appendix

## A1 AGNOSTOS Benchmark

### A1.1 Task Selection and Categorization

The AGNOSTOS benchmark is built upon the RLBench simulated environment, which includes 100 pre-defined manipulation tasks. Existing robotic manipulation studies typically use a subset of 18 RLBench tasks for training and close-set testing. Figure A1 illustrates these 18 tasks, which we refer to as seen tasks. To rigorously assess the generalization capabilities of vision-language-action (VLA) models on novel tasks, we select 23 unseen tasks from RLBench with distinct semantics compared to the seen tasks. Figure 1 visualizes these 23 unseen tasks. In the Supplementary Materials, we provide video examples of all seen and unseen tasks.

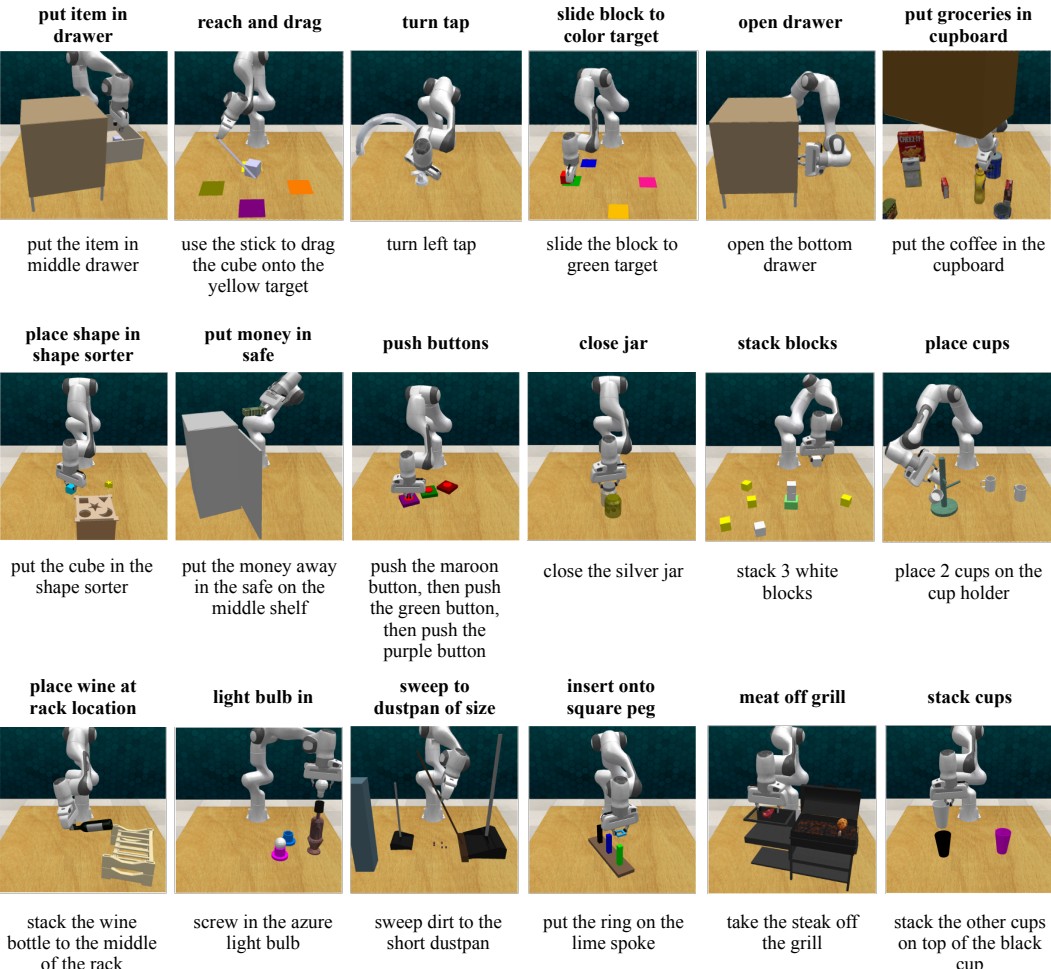

Figure A1: **Examples of 18 widely used training (seen) tasks on RLBench.**

The AGNOSTOS benchmark categorizes the 23 unseen tasks into two difficulty levels. The Level-1 testing includes 13 unseen tasks, which have similar semantics to seen tasks in either objects or motions. The Level-2 testing comprises 10 unseen tasks with entirely novel semantics, distinct from the seen tasks. The task categorization is performed by GPT-4o based on the semantics similarities between seen and unseen task descriptions. The categorization results are verified by three human experts in the field of robotics. In Table A1 we list all 23 unseen tasks, where the text in each "[]" symbol is an abbreviation for the corresponding task. Under the full task name of each unseen task, we show the seen task whose semantics are the most similar to the unseen task.

Table A1: 23 unseen tasks in our AGNOSTOS benchmark. The text in each "[]" symbol is an abbreviation for the corresponding task. Under the full task name of each unseen task, we show its most similar task among the 18 seen tasks.

| [Toilet]
put toilet roll on stand
(place wine at rack location) | [Knife]
put knife on
chopping board
(put item in drawer ) | [Fridge]
close fridge
(close jar) |
|---|---|---|
| [Microwave]
close microwave
(close jar) | [Laptop]
close laptop lid
(close jar) | [Phone]
phone on base
(place wine at rack location) |
| [Seat]
toilet seat down
(reach and drag) | [LampOff]
lamp off
(push buttons) | [LampOn]
lamp on
(push buttons) |
| [Book]
put books on bookshelf
(put groceries in cupboard) | [Umbrella]
put umbrella in
umbrella stand
(put item in drawer) | [Charger]
unplug charger
(no similar seen task) |
| [Grill]
open grill
(open drawer) | [Bin]
put rubbish in bin
(put item in drawer) | [USB]
take usb out of computer
(no similar seen task) |
| [Lid]
take lid off saucepan
(no similar seen task) | [Plate]
take plate off
colored dish rack
(no similar seen task) | [Ball]
basketball in hoop
(no similar seen task) |
| [Scoop]
scoop with spatula
(no similar seen task) | [Rope]
straighten rope
(no similar seen task) | [Oven]
turn oven on
(no similar seen task) |
| [Buzz]
beat the buzz
(no similar seen task) | [Plants]
water plants
(no similar seen task) | |

## A1.2 Evaluations of existing VLA models

Due to embodiment gaps (e.g., differences in robots, action spaces, and camera configurations), existing learning-based VLA models cannot effectively generalize to tasks in unseen embodiments. To address this, we fine-tune VLA models using data from RLBench's 18 seen tasks and evaluate their generalization on the 23 unseen tasks.

### A1.2.1 Human-video pre-trained VLA models

Existing VLA models pre-trained on large-scale human action videos, such as R3M [30], D4R [31], R3M-Align [32], and D4R-Align [32], claim to learn generalizable representations for robotic manipulation. HR-Align [32] adapts these models to RLBench by fine-tuning them on the 18 seen tasks, achieving strong performance on these tasks. We evaluate these adapted models provided by [32] on our 23 unseen tasks to assess their generalization.

### A1.2.2 In-domain data trained VLA models

For the VLA models trained on the RLBench's 18 training (seen) tasks (i.e., in-domain training), including PerAct [27], RVT [28], RVT2 [33], Sigma-Agent [78], and Instant Policy [69], we test their officially released models on our 23 unseen tasks. These VLA models serve as strong baselines on our benchmark, since they have sophisticated model designs on RLBench data and do not involve the data domain gap between their training tasks and our unseen testing tasks. Instant Policy is a within-task in-context imitation learning method that formulates policy prediction as a graph diffusion process over structured demonstrations and observations. To evaluate its cross-task generalization capability, we evaluate the released model on our 23 unseen tasks in a zero-shot manner, where a randomly sampled seen demonstration is used for in-context prompting.

### A1.2.3 Foundation VLA models

The foundation VLA models trained on large-scale robotic data or built upon LLM or VLM models, are expected to have strong generalization capabilities on new tasks. In this work, we evaluate Open-VLA [8], RDT [9], $\pi_0$ [1], LLARVA [77], SAM2Act [34], 3D-LOTUS++ [25], and VoxPoser [2]. Below, we elaborate on our fine-tuning details, which follow the official fine-tuning guidelines.

**OpenVLA [8].** We fine-tune OpenVLA using 3,600 demonstrations from the 18 seen tasks, with each demonstration comprising a front RGB view (size of $256 \times 256$) and the corresponding language instruction. We use a batch size of 16 and apply LoRA fine-tuning with a rank of 32 and a learning rate of $5 \times 10^{-4}$. Figure A2 shows the training loss and action accuracy during fine-tuning, indicating rapid convergence within 2,000 steps. We evaluate the model on the 23 unseen tasks every 1,000 steps and select the model with the highest generalization performance.

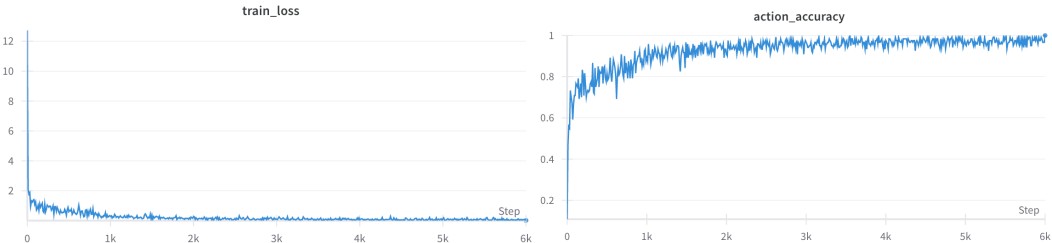

Figure A2: The statistics of the fine-tuning of OpenVLA.

**RDT [9].** For the fine-tuning of RDT, we also use the 3,600 seen demonstrations. For each demonstration, RDT takes the front and wrist RGB views as well as the language instruction as inputs, where the image size is $256 \times 256$. We use a batch size of 16 and fine-tune the RDT for 400,000 steps by following their official guidance. Figure A3 shows the training loss and the overall average sample MSE loss (a good metric claimed by the authors) during the fine-tuning, which indicates that the RDT converges well. We evaluate the RDT model every 10,000 steps, and report the highest generalization performance on our 23 unseen tasks.

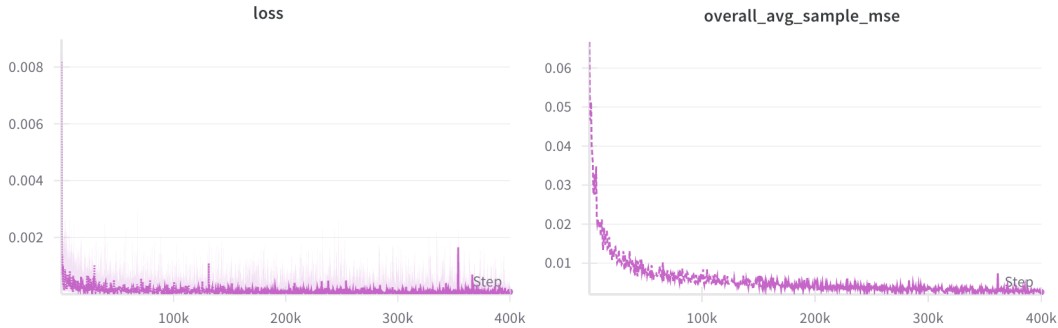

Figure A3: The statistics of the fine-tuning of RDT.

**$\pi_0$ [1].** We fine-tune $\pi_0$ with LoRA using 3,600 demonstrations, each including front, wrist, and overhead RGB views (size of $256 \times 256$) and the language instruction. Following the official protocol, we set the batch size to 64 and train for 100,000 steps. Figure A4 shows the training loss during fine-tuning. We evaluate the model every 10,000 steps and report the best generalization performance on the 23 unseen tasks.

**LLARVA [77].** LLARVA is a model trained with instruction tuning on the Open X-Embodiment dataset [29], which unifies various robotic tasks and environments. The authors further adapt LLARVA to RLBench to mitigate the embodiment gap. Therefore, we use the officially released model to evaluate the generalization performance on our 23 unseen tasks.

**3D-LOTUS/3D-LOTUS++ [25].** We directly evaluate the generalization performance of 3D-LOTUS using the pretrained model released by the authors. For 3D-LOTUS++, we follow the recommended configuration and adopt LLaMA3-8B as the LLM, and the base models of OwlViT v2 [89] and

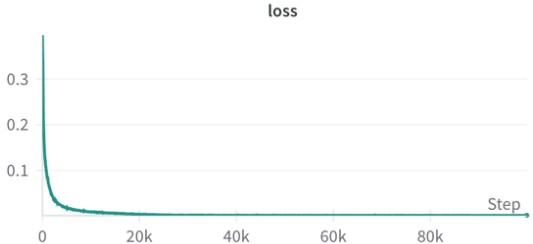

Figure A4: The statistics of the fine-tuning of $\pi_0$.

SAM [90] as the VLM components. These modules enable 3D-LOTUS++ to effectively decompose high-level instructions and ground object references in 3D space. We evaluate the model by following the official pipeline on our 23 unseen tasks.

**SAM2Act [34].** SAM2Act builds upon the RVT-2 framework and introduces a novel module that integrates semantic masks from SAM [90] into the action learning pipeline. The SAM2Act model is trained end-to-end on RLBench's 18 training tasks. In our experiment, we use the checkpoint released by the authors and evaluate its performance directly on our 23 unseen tasks.

**VoxPoser [2].** VoxPoser uses a large language model to generate 3D affordance maps for motion planning. We adopt the open-sourced Qwen2.5-72B-Instruct variant as the LLM and modify the original single-task pipeline to support batched evaluation on custom datasets. During evaluation, object positions are directly obtained from the simulator as ground-truth. We evaluate the model in a zero-shot manner on our 23 unseen tasks.

### A1.3   Performance on Seen Tasks.

In Table A2, we evaluate the performance of RVT2, R3M-Align, OpenVLA, $\pi_0$ and our X-ICM (72B) on 18 in-domain (seen) tasks, alongside their performance on 23 unseen tasks. These results demonstrate that all methods achieve high success rates on in-domain tasks after fine-tuning on seen task demonstrations, highlighting the challenge of cross-task zero-shot generalization.

For X-ICM, we use 18 within-task demonstrations to construct in-context prompts for in-domain tasks, yielding a 60.4% average success rate, which is substantially higher than its 30.1% on unseen tasks. However, our X-ICM's performance on in-domain tasks is lower than that of learning-based VLA methods like (70.5%). This gap is expected, as X-ICM relies solely on the LLM's in-context learning capability without fine-tuning.

|  | 18 seen/in-domain avg (std) | 23 unseen avg (std) |
|---|---|---|
| RVT2 | 82.3 (0.7) | 10.3 (0.6) |
| R3M-Align | 59.2 (0.8) | 12.2 (0.3) |
| OpenVLA | 63.1 (1.4) | 13.6 (0.8) |
| $\pi_0$ | 70.5 (0.9) | 17.5 (0.4) |
| X-ICM (72B) | 60.4 (0.7) | 30.1 (1.0) |

Table A2: Performance on 18 seen tasks and 23 unseen tasks.

## A2   More Analysis

### A2.1   Dynamics-guided Sample Selection module

**Different In-context Sample Selection Baselines.** Our Dynamics-guided Sample Selection module is designed to capture the dynamic semantics critical for cross-task generalization by modeling the transformation from an initial to a final state in manipulation tasks, as detailed in Section 4.2. Unlike standard methods like Video-CLIP [91], which rely on video sequences and are thus incompatible with our setting—where only a single initial frame is available during testing unseen tasks—our diffusion-based model predicts future observations from this single frame. This enables effective modeling of task temporal dynamics essential for generalization.

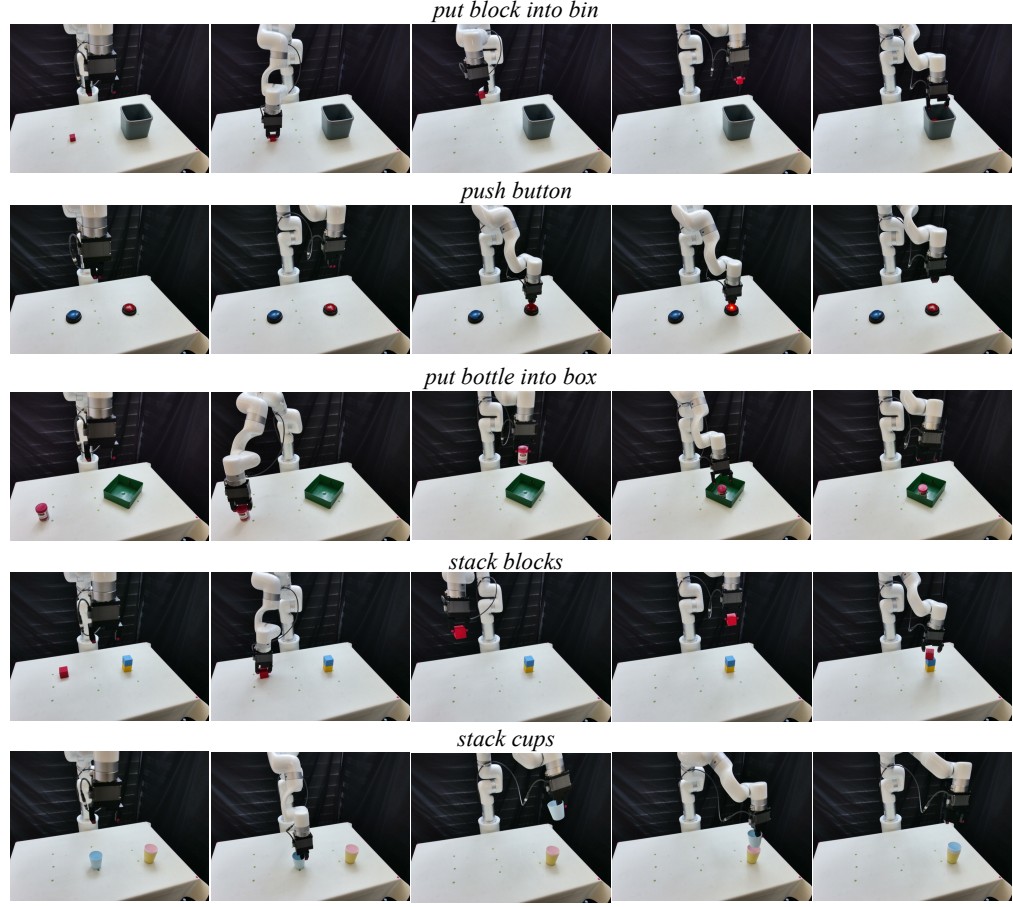

Figure A5: **Demonstration examples of five real-world tasks.** We visualize one demonstration for each task, where we show the key-observations captured by the third-view Orbbec camera.

To validate the necessity of our approach, we compared it with baselines using CLIP [92] and DINOv2 [93] features for sample selection, alongside a random selection strategy. The results in Table A3 demonstrate that our method achieves higher average success rates and lower variance compared to these general-purpose features, as it is trained specifically on manipulation dynamics, unlike the broader datasets used for CLIP and DINOv2.

|  | Level-1 avg (std) | Level-2 avg (std) | All avg (std) |
|---|---|---|---|
| Random | 30.7 (4.7) | 18.0 (2.2) | 25.2 (3.2) |
| CLIP feature | 32.1 (3.5) | 18.9 (2.6) | 26.4 (2.9) |
| DINOV2 feature | 33.7 (3.1) | 19.3 (2.0) | 27.4 (2.5) |
| Dynamics Diffusion (Ours) | 37.6 (1.4) | 20.3 (1.7) | 30.1 (1.0) |

Table A3: Comparisons between different feature representations.

**Qualitative results of the dynamics diffusion model.** To facilitate the cross-task sample selection, we train a dynamics diffusion model to encode the dynamic representations within each robot demonstration. The similarities of the captured dynamic representations across seen and unseen demonstrations are effective in identifying relevant demonstrations, thus guiding the cross-task sample selection process. For each demonstration, the dynamics diffusion model takes the initial observation and the corresponding language description as inputs, aiming to predict the final visual observation at the completion of the demonstration. Figure A6 showcases qualitative predictions from the trained dynamics diffusion model.

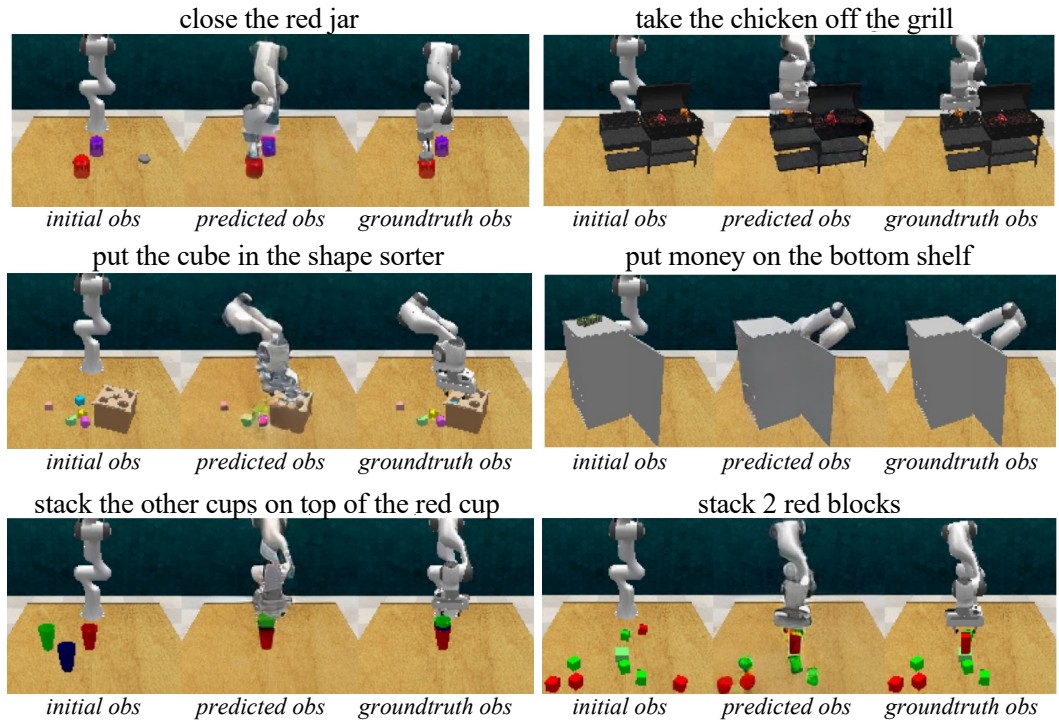

close the red jar | take the chicken off the grill

*initial obs  predicted obs  groundtruth obs*

put the cube in the shape sorter | put money on the bottom shelf

*initial obs  predicted obs  groundtruth obs*

stack the other cups on top of the red cup | stack 2 red blocks

*initial obs  predicted obs  groundtruth obs*

Figure A6: Qualitative results of the trained dynamics diffusion model. For each demonstration, the dynamics diffusion model uses the initial visual observation and language description as conditions and predicts the final visual observation upon task completion.

**Using different dynamic features.** Our dynamics diffusion model adopts the architecture of InstructPix2Pix [82]. After training, we extract multi-modal features from each demonstration to serve as dynamic features for sample selection. In this work, we use the combination of the textual feature of the language description (i.e., $feat_{lang}$) and the predicted latent feature of the target observation (i.e., $feat_{vis.out}$) as the dynamic features. Alternatively, we can use the latent feature of the initial observation (i.e., $feat_{vis.in}$), and try different combinations to form the dynamic features. Table A4 presents the generalization performance on the 23 unseen tasks for different feature combinations. The predicted latent feature of the target observation (i.e., $feat_{vis.out}$) proved most effective, validating the ability of our dynamics diffusion model to capture generalizable dynamics. Additionally, we find that the latent feature of the initial observation (i.e., $feat_{vis.in}$) is also effective, while using the textual feature of the language description (i.e., $feat_{lang}$) does not show benefit.

Table A4: Generalization performance using different dynamic feature combinations. Upon our InstructPix2Pix-based dynamics diffusion model, $feat_{lang}$ denotes the language instruction feature, $feat_{vis.in}$ represents the latent feature of the initial observation, and $feat_{vis.out}$ denotes the predicted latent feature of the target observation.

| $feat_{lang}$ | $feat_{vis.in}$ | $feat_{vis.out}$ | Level-1 | Level-2 | All |
|---|---|---|---|---|---|
| ✗ | ✗ | ✗ | 30.7 (4.7) | 18.0 (2.2) | 25.2 (3.2) |
| ✗ | ✓ | ✗ | 34.9 (2.9) | 17.8 (1.4) | 27.5 (1.0) |
| ✗ | ✗ | ✓ | 37.2 (2.4) | 19.9 (2.1) | 29.7 (2.2) |
| ✗ | ✓ | ✓ | 37.1 (0.8) | 17.6 (2.0) | 28.6 (1.3) |
| ✓ | ✓ | ✗ | 36.3 (0.9) | 16.4 (0.4) | 27.7 (0.7) |
| ✓ | ✗ | ✓ | 37.6 (1.4) | 20.3 (1.7) | **30.1 (1.0)** |
| ✓ | ✓ | ✓ | 38.6 (1.6) | 16.7 (2.4) | 29.0 (1.6) |

### A2.2 Different model sizes of LLMs.

We investigated the impact of model scale on performance by evaluating the X-ICM model at various sizes of Qwen2.5-Instruct: 7B, 14B, and 72B parameters. As shown in Table A5, larger models

generally tend to yield better performance. Performance improved noticeably as the model size increased from 7B to 14B. The performance gain diminished when scaling models to 72B. This suggests that while increasing model size is generally beneficial, the returns may diminish.

Table A5: Effects of different model sizes.

| Models | Level-1 | Level-2 | All |
|---|---|---|---|
| X-ICM (7B) | 28.6 (1.9) | 16.9 (1.3) | 23.5 (1.6) |
| X-ICM (14B) | 38.6 (0.5) | 18.1 (2.3) | 29.7 (0.7) |
| X-ICM (72B) | 37.6 (1.4) | 20.3 (1.7) | 30.1 (1.0) |

### A2.3    Cross-task In-context Prompt

Figure A7 shows an example of the construction process of the cross-task in-context prompt. The prompt is concatenated by the system prompt, textualized cross-task seen demonstrations that are selected, and the textualized tested unseen task.

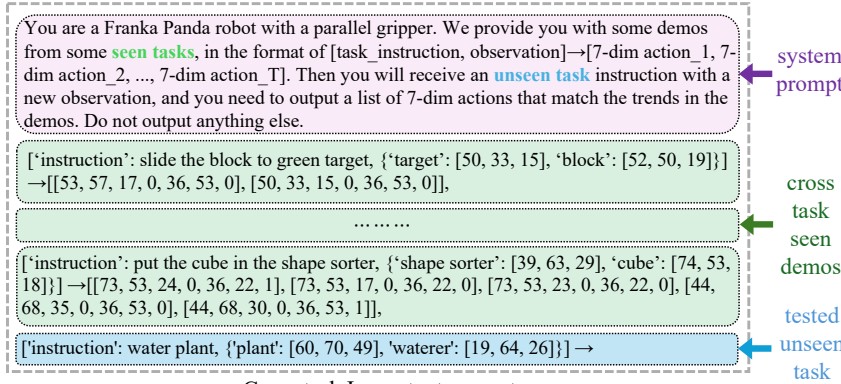

Cross-task In-context prompt

Figure A7: An example of the construction of our cross-task in-context prompt, which is concatenated by the system prompt, textualized cross-task seen demonstrations that are selected, and the textualized tested unseen task.

## A3    Real-world Manipulation Experiments

In Figure A5, we visualize the collected demonstrations for five real-world tasks. For each task, we randomly select one demonstration for visualization. And we visualize the visual observations when the key-actions occur in each demonstration. In the Supplementary Materials, we showcase some testing videos that include both successful and failed cases.

## A4    Broader Impacts Statements

There are no ethical issues involved in this paper. This work proposes a cross-task generalization benchmark to evaluate the generalization capabilities of existing vision-language-action models. The method proposed in this work is purely algorithmic. Therefore, the proposed benchmark and method does not change the societal impacts of robotic manipulation.

