# OpenReview forum: "Exploring the Limits of Vision-Language-Action Manipulation in Cross-task Generalization"
_NeurIPS.cc/2025/Conference — NeurIPS 2025 poster_

### Official Review · Reviewer_KVSd · 2025-06-30

**Clarity:** 3
**Significance:** 2
**Originality:** 2
**Rating:** 4
**Confidence:** 3

**Summary:**

This paper introduces AGNOSTOS, a new simulation benchmark built upon RLBench, which selects 23 unseen tasks as the test set and uses the standard 18-task configuration as the training set. To evaluate cross-task zero-shot generalization in manipulation, AGNOSTOS categorizes the unseen tasks into two levels based on their similarity to the training tasks. In addition to the benchmark, this paper proposes Cross-task In-Context Manipulation (X-ICM), a method specifically designed to tackle the challenges of cross-task zero-shot generalization. Compared to a broad range of mainstream Vision-Language Action (VLA) models, X-ICM achieves state-of-the-art performance on AGNOSTOS.

**Questions:**

1. Tasks such as straightening a rope or removing a USB device from a computer demand high spatial resolution, yet a single grid cell may encompass multiple parts from different objects, leading to ambiguity and reduced control precision. Could the authors explain further how the proposed X-ICM framework effectively handles such tasks under these discretization constraints.
2. Include a more detailed analysis of failure cases, distinguishing between issues related to sample selection and LLM reasoning. This could provide more insight for this paper.
3. The performance of X-ICM on complex, long-horizon tasks remains insufficiently explored, especially in realistic or real-world settings. Could the authors provide further experiment results in real-world to show X-ICM’s practical applicability?

**Ethical Concerns:**

["NO or VERY MINOR ethics concerns only"]

**Final Justification:**

Thanks for the detailed rebuttal. Most of my concerns and questions have been well addressed. I have raised my rating to BA. I'd further suggest the paper to include those experiments on long-horizon tasks under realistic settings in the revision.

**Limitations:**

Yes.

**Paper Formatting Concerns:**

No.

**Quality:**

2

**Strengths And Weaknesses:**

Strengths:
1. This paper proposes a new simulation benchmark, AGNOSTOS, to evaluate the cross-task zero-shot generalization in manipulation.
2. X-ICM shows effective and impressive experiment results, compared to other mainstream VLA methods, e.g. pi0 and OpenVLA.

Weaknesses:
1. The construction of AGNOSTOS appears relatively straightforward, as it involves a direct partitioning of existing tasks within RLBench. The novelty of this benchmark is thus somewhat unclear.
2. The methodological details of the proposed X-ICM approach lack sufficient clarity. For instance, it remains ambiguous how the output of the large language model (LLM) is transformed into precise low-level control commands, such as accurate gripper poses required for manipulation. A more detailed description of this mapping process is necessary to evaluate the feasibility and reproducibility of the method.
3. Voxposer should not be categorized as a Vision-Language Action (VLA) method. While it may share some similarities in using vision and language components, its core design and objectives differ significantly from mainstream VLA models.
4. There is a labeling error in Figure 2. where the language description in Diffusion-based dynamics modeling stage is incorrectly labeled as "Close Jar" instead of the correct "Close Box."

---

> ### Author Rebuttal · Authors · 2025-07-31
>
> **Dear Reviewer:**
>
> We greatly appreciate your recognition of our novel AGNOSTOS benchmark for evaluating cross-task zero-shot generalization and the strong performance of X-ICM compared to mainstream VLA methods. We believe that our clarifications below can address all your concerns and provide you with new perspectives to re-assess our paper’s contributions.
>
>
> &nbsp;
>
>
> **Q.1 Clarification on the construction of AGNOSTOS.**
>
> - We clarify the significant contributions of our AGNOSTOS benchmark to the VLA community:
>     - **Pioneering Cross-task Zero-shot Benchmark:** AGNOSTOS is the first systematic benchmark designed for cross-task zero-shot manipulation, addressing a critical gap in the literature where prior work has largely overlooked this challenging setting.
>     - **Revealing Generalization Limitations:** Through extensive evaluation of multiple VLA methods on AGNOSTOS, we found that existing methods achieve success rates below 20% on 23 unseen tasks (Table 2). This insight highlights the limitations of current approaches and provides a standardized framework to guide future VLA generalization research.
> - **The decision to build upon RLBench was deliberate**: RLBench is a widely adopted, high-fidelity manipulation environment used by influential works (e.g., PerAct [27], RVT2 [33]) and generalization benchmarks (e.g., COLOSSEUM [24], GemBench [25]). Leveraging RLBench ensures informative and reliable comparisons with prior methods, maintaining a cohesive ecosystem for the manipulation community rather than introducing a new simulation environment.
>
>
> &nbsp;
>
>
> **Q.2 Transformation between LLM’s input/output and low-level actions.**
>
> We provide a detailed description of the transformation process below.
>
> We define the robot’s workspace in the base coordinate system with ranges $X: [-0.3m, 0.7m]$, $Y: [-0.5m, 0.5m]$, and $Z: [0.6m, 1.6m]$. An end-effector action at a given timestamp is represented as a 7-dimensional vector, e.g., $[0.1m, -0.2m, 0.8m, 20°, 90°, 240°, 1]$, where the first three elements denote the 3D position (X, Y, Z), the next three denote the rotation angles around the X-, Y-, and Z-axes, and the last element indicates the gripper state (0 for closed, 1 for open).
>
> We discretize the workspace and angles as follows:
> - The 3D workspace is divided into 100 uniform grids along each axis. For a position $p$ in range $[p_{\text{min}}, p_{\text{max}}]$, the discretized value is computed as: $$\text{Grid} = \left\lfloor \frac{p - p_{\text{min}}}{p_{\text{max}} - p_{\text{min}}} \times 100 \right\rfloor$$
> - Each rotation angle is discretized into the range $[0, 72]$ with a $5°$ interval, yielding 72 bins. The discretized angle is: $$\text{Bin} = \left\lfloor \frac{\text{angle}}{5} \right\rfloor$$
> - The gripper state remains unchanged (0 or 1).
>
> For the example action $[0.1m, -0.2m, 0.8m, 20°, 90°, 240°, 1]$, we convert it into LLM's input format:
> - X-position: $\frac{0.1 - (-0.3)}{0.7 - (-0.3)} \times 100 = 40$
> - Y-position: $\frac{-0.2 - (-0.5)}{0.5 - (-0.5)} \times 100 = 30$
> - Z-position: $\frac{0.8 - 0.6}{1.6 - 0.6} \times 100 = 20$
> - X-rotation: $\frac{20}{5} = 4$
> - Y-rotation: $\frac{90}{5} = 18$
> - Z-rotation: $\frac{240}{5} = 48$
> - Gripper state: $1$
>
> Thus, the action in LLM format is $[40, 30, 20, 4, 18, 48, 1]$. The inverse process can convert the action in LLM's format into continuous end-effector action.
>
> We will incorporate these details into the revised manuscript to enhance clarity.
>
> &nbsp;
>
> **Q.3 Clarification on the Voxposer method.**
>
> We clarify that since much of the literature (eg, [R3][R4]) in the community classifies Voxposer as a VLA method, we follow this classification. We will give a special explanation of the difference between Voxposer and mainstream VLA in the paper. Thank you for pointing this out.
>
> [R3] A Survey on Vision-Language-Action Models: An Action Tokenization Perspective, 2025.
>
> [R4] Vision-Language-Action Models: Concepts, Progress, Applications and Challenges, 2025
>
> &nbsp;
>
> **Q.4 Labeling error in Figure 2.**
>
> Thank you, we will correct it.
>
> &nbsp;
>
> **Q.5 Clarification on the spatial perception with discretization constraints.**
>
> As detailed in our response to Q.2, we discretize the robot’s workspace into $1 cm^2$ grids (100 grids per axis across X: [-0.3m, 0.7m], Y: [-0.5m, 0.5m], Z: [0.6m, 1.6m]). Given the sparse and scattered object distribution in the RLBench environment, the likelihood of a single $1 cm^2$ grid encompassing multiple object parts is minimal, reducing ambiguity in most cases.
>
> For the "USB" tasks, the success rate is very high, and we found that this is because the detection of "USB" is very accurate. For the "straight rope" task, the detection performance for the "rope head" is poor, which is the main reason for its low success rate.
> Therefore, we hope that future work can focus on enhancing the 3D spatial understanding ability of models.
>
> &nbsp;
>
> **Q.6 Failure analysis (sample selection vs. LLM reasoning).**
>
> Thank you for this interesting and valuable question. In order to investigate the impact of sample selection and LLM reasoning on model performance, we compared two experimental settings. One is our original X-ICM model, which follows a cross-task zero-shot setting. Another one is that in the testing of unseen tasks, we do not apply the cross-task sample selection, but instead provide a single sample within the unseen tasks, so that we can probe the performance of X-ICM under the one-shot setting.
>
> On each task, by comparing the performance of the model in the cross-task zero-shot setting (both sample selection and LLM reasoning are involved) and the within-task one-shot setting (providing the most favorable within-task sample, LLM reasoning is more involved), we can perform the following analysis and obtain conclusions:
> - If the model performs equally in the two situations (eg, within 5% performance difference), it means that LLM's reasoning ability has a greater impact on the task;
> - If the within-task one-shot setting is significantly better (eg, more than 5% performance difference) than the cross-task zero-shot setting, it means that sample selection has a greater impact on the task;
> - If the cross-task zero-shot setting is significantly better (eg, more than 5% performance difference) than the within-task one-shot setting, it means that LLM's reasoning ability has a greater impact on the task;
>
> In the table below, we compare the model's performance in these two situations. The table follows the same format as Table 2 in our paper (for simplicity, we use "T(task_id)" to represent the original task names in Table 2). From the results, we can see that, for tasks "T3, T7, T9, T10, T12, T13, T15, T17, T19, T20, T22, T23", the sample selection is more important; For the remaining tasks, the LLM reasoning is more important.
>
>
> | - | T1 | T2 | T3 | T4 | T5 | T6 | T7 | T8 | T9 | T10 | T11 | T12 | T13 |
> | :---: | :---: | :---: | :---: | :---: | :---: | :---: | :---: | :---: | :---: | :---: | :---: | :---: | :---: |
> | X-ICM (72B, **zero-shot**) | 6.7 | 69.3 | 12.7 | 58.7 | 34.0 | 68.0 | 51.3 | 86.7 | 74.7 | 2.0 | 1.3 | 5.3 | 18.7 |
> | X-ICM (72B, **one-shot**) | 0.0 | 58.7 | 57.3 | 33.3 | 33.3 | 61.3 | 69.3 | 70.7 | 97.3 | 10.7 | 0.0 | 57.3 | 72.0 |
>
> | - | T14 | T15 | T16 | T17 | T18 | T19 | T20 | T21 | T22 | T23 | **Level-1 avg (std)** | **Level-2 avg (std)** | **All avg (std)** |
> | :---: | :---: | :---: | :---: | :---: | :---: | :---: | :---: | :---: | :---: | :---: | :---: | :---: | :---: |
> | X-ICM (72B, **zero-shot**) | 98.7 | 13.3 | 4.7 | 36.0 | 0.7 | 16.0 | 20.7 | 7.3 | 2.7 | 2.7 | 37.6 (1.4) | 20.3 (1.7) | 30.1 (1.0) |
> | X-ICM (72B, **one-shot**) | 98.7 | 94.7 | 8.0 | 52.0 | 5.3 | 28.0 | 50.7 | 4.0 | 8.0 | 14.7 | 47.8 (0.8) | 36.4 (1.9) | 42.8 (0.7) |
>
>
> &nbsp;
>
>
> **Q.7 Clarification on long-horizon task setting.**
>
> Thank you for your suggestion to explore X-ICM’s performance on complex, long-horizon tasks in realistic settings. To clarify, our work focuses on the first systematic evaluation of cross-task zero-shot generalization for VLA methods, a challenging and under-explored problem. Learning dynamic semantic generalization across tasks from robot data is inherently difficult, as VLM foundation models lack this capability. Even simple unseen tasks pose significant challenges in this setting, and long-horizon tasks, requiring sequential sub-task execution, amplify these difficulties. We believe these tasks warrant further in-depth study.
>
> At your request, we explore this interesting problem in real-world experiments. We use tasks *"put block into bin"*, *"push button"*, *"put bottle into box"*, *"stack blocks*", and *"pull out the middle block"* as seen tasks, where we have 10 demonstrations for each task.
>
> **The long-horizon task we evaluated is *"clean the table"***, according the robot's visual observation, we decompose the long-horizon task into three sequential sub-tasks, ie, *"stack cups"*, *"put the stacked cups into plate"*, *"place the mango on top of the stacked cups"*. We evaluated this task over 20 rollouts, with sub-task success contingent on the success of all prior sub-tasks.
>
> For our X-ICM method, the success rates on sub-tasks *"stack cups"*, *"put the stacked cups into plate"*, *"place the mango on top of the stacked cups."* are 40%, 25%, and 5%, respectively. Thus, the overall success rate for the long-horizon task is 5%.
>
> For comparison, we extend the RoboPrompt (within-task in-context learning) method to our cross-task zero-shot setting by randomly selecting cross-task samples. And the success rates on the three sub-tasks are 10%, 0%, and 0%, respectively. This also demonstrates the effectiveness of our method.
>
> These results align with expectations, as the sequential nature of long-horizon tasks leads to cumulative failure. The cross-task zero-shot setting exacerbates these difficulties. These findings highlight the need for more advanced algorithms to handle long-horizon tasks in zero-shot scenarios.

---

> > ### Comment · Reviewer_KVSd · 2025-08-07
> >
> > Thanks for the detailed rebuttal. Most of my concerns and questions have been well addressed. I have raised my rating to BA. I'd further suggest the paper to include those experiments on long-horizon tasks under realistic settings in the revision.

---

> > > ### Author Response · Authors · 2025-08-07
> > >
> > > Dear Reviewer KVSd,
> > >
> > > Thank you very much for reading our responses. We are pleased to hear that our responses have addressed most of your concerns. We will include the experiments on cross-task zero-shot long-horizon tasks provided in the responses in our revision.
> > >
> > > Thank you again for your time and effort.
> > >
> > > Sincerely,
> > >
> > > The Authors.

---

### Official Review · Reviewer_Xb1Z · 2025-07-01

**Clarity:** 3
**Significance:** 3
**Originality:** 3
**Rating:** 4
**Confidence:** 5

**Summary:**

In this work, the authors present AGNOSTOS, a benchmark designed to evaluate cross-task zero-shot generalization for robotic manipulation. The analysis of various VLA models reveals significant limitations in generalizing to unseen tasks. To address this challenge, the authors introduce X-ICM, a cross-task in-context manipulation approach that harnesses the generalization capabilities of LLMs, achieving significant improvements over existing VLA models.

**Questions:**

1) **Comparison to RoboPrompt and KAT.** While I agree that existing in-context learning baselines such as RoboPrompt and KAT are not designed for cross-task zero-shot generalization scenarios, including them in the comparison would still provide valuable insight into their current performance.


2) **“Dynamics-guided sample selection” Ablations.** This section lacks clarity. It would be helpful to evaluate multiple approaches or baselines to highlight the challenges in selecting effective in-context examples. The authors seem to have chosen diffusion models on purpose, but the reason for this choice is not clearly explained. Simpler methods could be considered, so the authors are encouraged to provide further details, motivations, and comparisons with alternative approaches that were tested but found less effective.


3) The constructed prompt closely resembles the one used in RoboPrompt. If this is accurate, please ensure proper citation within lines 225-235.


Overall, I liked this paper. However, it would benefit from additional justification for the proper use of X-ICM and a more thorough comparison to closely related baselines like KAT/RoboPrompt.

**Ethical Concerns:**

["NO or VERY MINOR ethics concerns only"]

**Final Justification:**

After reading the authors' feedback and other reviewers' opinions, I would like to thank the authors for their rebuttal.

My questions have been fully addressed. Therefore, I am maintaining my score for acceptance, as this paper explores cross-task settings for in-context learning in robotics, which I consider an important direction for future research.

**Limitations:**

yes

**Quality:**

3

**Strengths And Weaknesses:**

1) In general, the paper is clear and the technical details are easy to understand.


2) The core idea of cross-task zero-shot generalization in vision-language-action (VLA) models and introducing a benchmark for this purpose is a valuable and timely contribution to the field.


3) The related work section is thorough and extensive.

---

> ### Author Rebuttal · Authors · 2025-07-31
>
> **Dear Reviewer:**
>
> We sincerely thank you for recognizing the clarity of our paper and the valuable contribution of our AGNOSTOS benchmark in advancing cross-task zero-shot generalization for vision-language-action models. We believe that our clarifications below can address all your concerns.
>
> &nbsp;
>
>
> **Q.1 Comparison to RoboPrompt and KAT**
>
> Since Roboprompt and KAT are designed for within-task in-context learning and rely on random demonstration selection, we extend them to our cross-task setting by randomly selecting cross-task samples. To ensure a fair comparison, all methods use the Qwen2.5-72B-Instruct model as the LLM backbone.
> The results, shown below, demonstrate that X-ICM significantly outperforms KAT and RoboPrompt in both average success rate and stability, owing to our dynamics-guided cross-task sample selection module.
>
> | - | **Level-1 avg (std)** | **Level-2 avg (std)** | **All avg (std)** |
> | :---: | :---: | :---: | :---: |
> | **KAT** | 24.8 (5.8) | 15.3 (4.4) | 20.7 (4.7) |
> | **Roboprompt** | 30.7 (4.7) | 18.0 (2.2) | 25.2 (3.2) |
> | **X-ICM (Ours)** | **37.6 (1.4)** | **20.3 (1.7)** | **30.1 (1.0)** |
>
>
> &nbsp;
>
>
> **Q.2 Dynamics-guided sample selection Ablations.**
>
> We clarify the motivation for our module design and provide additional baseline comparisons to highlight its effectiveness.
>
> **(1) Motivation for Dynamics-guided Sample Selection:** Our work tackles the challenging problem of cross-task zero-shot manipulation via in-context learning, where selecting relevant cross-task demonstrations is critical. Effective sample selection requires modeling the temporal dynamics inherent in manipulation tasks. Given that only an initial RGB frame and task description are available during testing, we train a diffusion model to predict the final visual observation from the initial observation. This approach enables the model to learn feature representations that encode task temporal dynamics, facilitating robust cross-task sample selection.
>
> **(2) Ablation with Additional Baselines:** To demonstrate the effectiveness of our dynamics-guided sample selection, we compared it against two general-purpose feature extractors, CLIP and DINOv2, and a random selection baseline. The results, shown below, indicate that CLIP and DINOv2 outperform random selection but are surpassed by our dynamics diffusion model, which achieves higher performance and stability due to its focus on task dynamics.
>
> | - | **Level-1 avg (std)** | **Level-2 avg (std)** | **All avg (std)** |
> | :---: | :---: | :---: | :---: |
> | **Random** | 30.7 (4.7) | 18.0 (2.2) | 25.2 (3.2) |
> | **CLIP feature** | 32.1 (3.5) | 18.9 (2.6) | 26.4 (2.9) |
> | **DINOv2 feature** | 33.7 (3.1) | 19.3 (2.0) | 27.4 (2.5) |
> | **Dynamics Diffusion (Ours)** | **37.6 (1.4)** | **20.3 (1.7)** | **30.1 (1.0)** |
>
> &nbsp;
>
>
> **Q.3 Clarification on the prompt design.**
>
> Thank you for your feedback regarding the prompt. To clarify, as our method focuses on cross-task zero-shot generalization rather than prompt engineering, we adapt the prompt structure from RoboPrompt to suit our cross-task setting. We will ensure proper citation of RoboPrompt in lines 225-235 of the revised manuscript to acknowledge this source. Thank you for bringing this to our attention.

---

> > ### Comment · Reviewer_Xb1Z · 2025-08-05
> > **Response**
> >
> > After reading the authors' feedback and other reviewers' opinions, I would like to thank the authors for their rebuttal.
> >
> > My questions have been fully addressed. Therefore, I am maintaining my score, as this paper explores cross-task settings for in-context learning in robotics, which I consider an important direction for future research.

---

> > > ### Author Response · Authors · 2025-08-05
> > >
> > > Dear Reviewer Xb1Z,
> > >
> > > Thank you for taking the time to read our rebuttal. We are pleased to hear that our response has fully addressed your questions. We truly appreciate your recognition of our work, which aims to push the limits of cross-task generalization in the existing VLA community. Your encouragement is very important to us.
> > >
> > > Thank you once again for your valuable time and support.
> > >
> > > Sincerely,
> > >
> > > The Authors

---

### Official Review · Reviewer_Qo7p · 2025-07-02

**Clarity:** 3
**Significance:** 2
**Originality:** 2
**Rating:** 3
**Confidence:** 4

**Summary:**

This paper introduces AGNOSTOS, a simulation benchmark built upon RL-Bench, which includes 23 unseen manipulation tasks for evaluation. By testing several existing Vision-Language-Action (VLA) models on this benchmark, the authors observe that these models tend to generalize well to seen training tasks but perform poorly on unseen tasks. The authors propose a novel method called Cross-Task In-Context Manipulation (X-ICM) to address this limitation. This approach leverages large language models (LLMs) to predict actions by prompting them to demonstrate similar seen tasks used as in-context examples, which are selected via a dynamics-guided sampling strategy. This dynamics-guided sampling strategy first uses an image editing model based on InstructPix2Pix trained using the demonstration database of seen tasks to map (text, image) pairs to goal images, and then uses features derived from this model to compute similarity and retrieve the top-k most relevant demonstrations. On the proposed AGNOSTOS benchmark, the X-ICM method outperforms prominent VLA baselines such as $\pi_0$ and VoxPoser, demonstrating its effectiveness in generalizing to unseen tasks.

**Questions:**

1. Why do the authors choose to use a diffusion model trained on the simulation benchmark dataset for feature extraction, rather than adopting a more generalizable Video-CLIP-based method? The former approach incurs additional training costs and lacks transferability to other benchmarks beyond RL-Bench, undermining the claimed generality of the proposed X-ICM framework.
2. What is the success rate of the X-ICM framework without providing any demonstrations? This baseline is important to quantify the actual contribution of the in-context prompting. Additionally, why not consider training an end-to-end VLA model that supports in-context learning, such as ICRT [4], which is capable of acting without demonstrations and improves performance when examples are provided?
3. The authors are encouraged to clarify their definition of cross-task zero-shot. As discussed earlier, the current formulation—using retrieved demonstrations from semantically similar tasks—does not strictly align with the conventional understanding of zero-shot generalization.

[4] In-Context Imitation Learning via Next-Token Prediction

**Ethical Concerns:**

["NO or VERY MINOR ethics concerns only"]

**Final Justification:**

Thanks for the comprehensive rebuttal, which addresses most of my concerns. I would like to raise my score to borderline.

**Limitations:**

Yes

**Paper Formatting Concerns:**

No major formatting issue.

**Quality:**

2

**Strengths And Weaknesses:**

=== Strengths ===
1. The paper presents some real-world experiments beyond simulation benchmarks, which makes the effectiveness of the proposed methods more convincing.
2. The authors present a well-motivated contribution by proposing a novel simulation benchmark named AGNOSTOS. This benchmark not only reveals that current VLA models fail to generalize effectively to unseen tasks but also provides a playground for future research on cross-task generalization in VLA models.
3. Leveraging in-context learning and generalized features of LLM, the proposed X-ICM can perform better than diverse VLA models on AGNOSTOS, showcasing its powerful cross-task zero-shot capability.

=== Weakness ===
1. The Dynamics-guided Sample Selection module appears relatively complex, and its motivation is not justified. In the current literature on multimodal RAG, especially video retrieval (e.g., VideoAgent [1], MineDreamer [2]), it is common to use more generalizable Video-CLIP-based methods to extract features and compute similarity for retrieving relevant video demonstrations. In contrast, this paper trains an image-editing model (based on data from RL-Bench) specifically for feature extraction, which may limit generalization. This approach seems misaligned with the paper’s stated goal of cross-task zero-shot generalization. Specifically, the proposed framework may not be easily transferable to new benchmarks or environments without retraining the diffusion-based image-editing model, thus undermining the claimed generality of X-ICM.
2. The proposed X-ICM method lacks novelty, as its within-task in-context manipulation approach is not substantially different from prior work (e.g., RoboPrompt[3]). Both methods retrieve relevant video demonstrations and then pass key annotation information to an LLM or VLM to directly generate actions, limiting the contribution of X-ICM from a methodological view.
3. The definition of zero-shot in this work is somewhat ambiguous. The X-ICM framework retrieves several demonstrations as contextual prompts for the LLM, even though these demonstrations may differ in actions/motions and objects from the target task (especially in Level 1, where partial semantic overlap exists). However, using such retrieved demonstrations as in-context examples deviates from the conventional definition of zero-shot, and may lead to confusion.

[1] VideoAgent: A Memory-augmented Multimodal Agent for Video Understanding

[2] Minedreamer: Learning to follow instructions via chain-of-imagination for simulated-world control

[3] In-context learning enables robot action prediction in LLMs

---

> ### Author Rebuttal · Authors · 2025-07-31
>
> **Dear Reviewer:**
>
> We greatly appreciate your recognition of our sufficient experiments, well-motivated and significant benchmark, and the strong performance. We believe that our clarifications below can address all your concerns and provide you with new perspectives to re-assess our paper’s contributions.
>
> &nbsp;
>
>
> **Q.1 Clarification on Dynamics-guided Sample Selection.**
>
> **(1) Clarification on the motivation:**
>
> Our Dynamics-guided Sample Selection module is designed to capture the dynamic semantics critical for cross-task generalization by modeling the transformation from an initial to a final state in manipulation tasks, as detailed in Section 4.2. Unlike standard methods like Video-CLIP, which rely on video sequences and are thus incompatible with our setting—where only a single initial frame is available during testing unseen tasks—our diffusion-based model predicts future observations from this single frame. This enables effective modeling of task temporal dynamics essential for generalization.
>
> To validate the necessity of our approach, we compared it with baselines using CLIP and DINOv2 features for sample selection, alongside a random selection strategy. The results, shown below, demonstrate that our method achieves higher average success rates and lower variance compared to these general-purpose features, as it is trained specifically on manipulation dynamics, unlike the broader datasets used for CLIP and DINOv2.
>
> | - | **Level-1 avg (std)** | **Level-2 avg (std)** | **All avg (std)** |
> | :---: | :---: | :---: | :---: |
> | **Random** | 30.7 (4.7) | 18.0 (2.2) | 25.2 (3.2) |
> | **CLIP feature** | 32.1 (3.5) | 18.9 (2.6) | 26.4 (2.9) |
> | **DINOv2 feature** | 33.7 (3.1) | 19.3 (2.0) | 27.4 (2.5) |
> | **Dynamics Diffusion (Ours)** | **37.6 (1.4)** | **20.3 (1.7)** | **30.1 (1.0)** |
>
> We will incorporate this analysis and ablation table into the revised manuscript to clarify the motivation and effectiveness of our approach.
>
>
> **(2) Clarification on cross-task generalization:**
>
> Regarding the generalizability of the X-ICM framework, we would like to clarify the scope of generalization we address. Due to the "embodiment gap" (e.g., a new robot or environment), virtually all learning-based VLA models require fine-tuning to adapt to a new data domain. Our work therefore focuses on a model's ability to generalize to unseen tasks within the same domain after this initial adaptation. Therefore, we argue that it is acceptable to train our X-ICM model on seen task data.
>
> In addition, for direct transfer to a new benchmark without training, our dynamics diffusion model can be replaced with an off-the-shelf feature extractor like DINOv2. As shown in the ablation in our response to the previous question, this approach outperforms random selection, confirming the inherent effectiveness and transferability of our cross-task sample selection mechanism itself.
>
> &nbsp;
>
>
> **Q.2 Comparison between our X-ICM and Roboprompt.**
>
> We clarify the key distinctions and novel contributions of our X-ICM method compared to prior work like RoboPrompt: while both leverage in-context learning, they address fundamentally different problems, and our method introduces a crucial, novel component as a result. Specifically,
>
> **A New Problem Setting:** RoboPrompt focuses on within-task generalization—learning to perform a seen task in a new scenario. Our work is the first to systematically apply and demonstrate the effectiveness of this in-context learning paradigm for the more challenging cross-task zero-shot setting. As shown in our main results (Table 2), this approach significantly outperforms existing VLA models on unseen tasks.
>
> **A Critical Selection Strategy:** In a within-task setting, randomly selecting demonstrations is a reasonable strategy, as all examples are relevant. However, in the cross-task setting, selecting relevant examples from a diverse pool of different tasks is non-trivial and critical for success. RoboPrompt does not address this challenge. Our core methodological contribution is the Dynamics-guided Sample Selection module, which is specifically designed to solve this problem by identifying dynamically similar demonstrations across different tasks. The following ablation demonstrates that applying a random selection strategy (analogous to RoboPrompt's approach) in our cross-task setting yields poor and unstable results. Our dynamics-guided selection provides a substantial performance gain, highlighting its necessity and novelty.
>
> | - | **Level-1 avg (std)** | **Level-2 avg (std)** | **All avg (std)** |
> | :---: | :---: | :---: | :---: |
> | **Random** | 30.7 (4.7) | 18.0 (2.2) | 25.2 (3.2) |
> | **Dynamics Diffusion (Ours)** | **37.6 (1.4)** | **20.3 (1.7)** | **30.1 (1.0)** |
>
>
> &nbsp;
>
>
> **Q.3 Clarification on the zero-shot definition.**
>
> We clarify that our definition of zero-shot learning in the X-ICM framework is both rigorous and consistent with established standards in the literature [R1][R2]. Our 23 unseen tasks are divided into two difficulty levels: Level-1 tasks share partial similarity with seen tasks in either objects or motions (unseen object or unseen motion settings), while Level-2 tasks have no similarity, representing unseen object-motion compositions.
>
> For Level-1 tasks, the partial dissimilarity (unseen objects or motions) aligns with the conventional zero-shot learning paradigm, as supported by the zero-shot human-object interaction (HOI) detection literature, which is closely related to our setting. Specifically:
>
> - CLIP4HOI (NeurIPS 2023) [R1] defines zero-shot HOI detection to include unseen object (UO), unseen verb (UV), and unseen composition (UC) settings, as noted in the caption of Table 1: “Zero-shot HOI detection results on HICO-DET. UC, UO, and UV denote unseen composition, unseen object, and unseen verb settings.” This directly supports our inclusion of unseen object and motion settings in Level-1 tasks.
> - KI2HOI [R2] similarly categorizes zero-shot HOI detection into three scenarios: unseen object, unseen action, and unseen combination, as stated in Section II.C (page 2): “Zero-shot HOI detection focuses on transferring knowledge from known HOI concepts to unseen classes. They can be categorized into three scenarios: unseen object, unseen action, and unseen combination.”
>
> [R1] CLIP4HOI: Towards Adapting CLIP for Practical Zero-Shot HOI Detection, NeurIPS 2023.
>
> [R2] Towards Zero-shot Human-Object Interaction Detection via Vision-Language Integration, Neural Networks, 2025.
>
>
> &nbsp;
>
> **Q.4 Performance without any demonstrations.**
>
> We conducted the zero-demonstration experiment and we found that the success rates on all 23 unseen tasks are 0%. This result is expected. Without any in-context examples, the LLM has no reference to follow the action patterns required to complete the manipulation tasks.
>
>
> &nbsp;
>
>
> **Q.5 Clarification on end-to-end in-context learning.**
>
> Thank you for suggesting an end-to-end training model like ICRT that supports in-context learning. To clarify, ICRT is designed for within-task in-context learning, where it leverages demonstrations to adapt to new scenarios of a seen task. Extending ICRT to our cross-task zero-shot setting, where models must generalize to entirely unseen tasks, requires significant further investigation, as this problem introduces unique challenges in sample selection and task dynamics transfer, as discussed in Sections 4.2 and 4.3.
>
> Additionally, our results (Table 2) show that strong cross-task performance relies on a powerful VLM backbone (e.g., VLM-based $\pi_0$ performs well while non-VLM-based RVT2 performs badly). However, training an end-to-end VLM-based model for in-context learning, such as adapting ICRT, incurs prohibitive computational costs due to the need for large-scale, diverse training data.

---

> ### Comment · Reviewer_Qo7p · 2025-08-05
>
> I appreciate the authors’ substantial efforts in the rebuttal and the additional experiments, which addressed many of my concerns.  In particular, the clarification regarding the motivation for dynamics-guided sample selection - necessitated by access to only the task’s initial frame - is helpful, as well as is the detailed explanation of the zero-shot definition.
>
> However, a few issues remain after reading the response:
>
> 1. Regarding cross-task generalization (Q.1(2)), the authors clarify that their scope focuses on learning-based VLA models, which require fine-tuning for new data domains. However, both Line 107 (“prompt LLMs for action generation”) and Figure 2 involve pre-trained LLMs generating actions in text form, which are not learning-based VLA models. This creates confusion: while the paper identifies limitations in learning-based VLA models, the proposed solution relies on pre-trained LLMs to generate action rather than directly addressing or improving those learning-based VLA models, which appear misaligned with the stated motivation.
>
> 2. In addressing cross-task generalization, the authors mention that their dynamics diffusion model can be replaced by an off-the-shelf feature extractor like DINOv2 to enable transfer without training. While dynamics diffusion achieves better performance, it introduces additional training overhead for adaptation to new domains. Although the authors assume initial training on seen tasks is acceptable, this requirement somewhat weakens the contribution.
>
> I would raise my score to borderline, but a little bit towards rejection. If the authors can further clarify these points, I may consider a higher rating.

---

> ### Author Response · Authors · 2025-08-05
> **Further Clarifications**
>
> Dear Reviewer Qo7p,
>
> Thank you for your detailed feedback on our rebuttal. We value your effort and are pleased that our clarifications on dynamics-guided sample selection and zero-shot definition addressed many concerns. Below, we respond to your remaining comments on generalization, clarifying the X-ICM framework’s alignment with our paper’s objectives and addressing potential misunderstandings respectfully.
>
> &nbsp;
>
> **[Discussion] Issue #1: Clarification on Q1(2) and Misunderstandings.**
>
> (1). Misunderstanding of Q1(2):
>
> We believe Issue #1 arises from a misinterpretation of our response to Q1(2). To resolve this, we first provide a more detailed explanation of the original question. The question Q1(2) is: “Specifically, the proposed framework may not be easily transferable to new benchmarks or environments without retraining the diffusion-based image-editing model, thus undermining the claimed generality of X-ICM.”
>
> Detailed Explanation: For the new benchmarks or environments, they are actually new embodiments, **so this is actually a cross-embodiment generalization problem** (ie, enabling a model to generalize to new embodiments without additional training). However, **our work does not focus on cross-embodiment generalization**, which is a well-recognized, unresolved challenge in the VLA community.
>
> (2). Scope of Our Work:
>
> The primary focus of our paper is to develop a benchmark for evaluating the **cross-task generalization** capabilities of various VLA methods. As demonstrated in Table 2, our evaluation encompasses **both learning-based VLA methods (e.g., OpenVLA, $\pi_0$) and learning-free VLA methods (eg, VoxPoser)**. We specifically assess **cross-task generalization within a single embodiment.**
>
> The reviewer’s comments in Issue #1, e.g., “Regarding cross-task generalization (Q.1(2)), the authors clarify that their scope focuses on learning-based VLA models, which require fine-tuning for new data domains” and “while the paper identifies limitations in learning-based VLA models,” suggest a misunderstanding. We did not intend to express that the scope of our work on the problem of cross-task generalization is limited to learn-based VLA methods. In fact, our evaluation included both learning-based methods (e.g., OpenVLA, Pi0) and learning-free methods (e.g., VoxPoser, RoboPrompt[see response to Q1 of reviewer Xb1Z]). Similarly, the “limitations” mentioned refer to cross-embodiment generalization, not cross-task generalization.
>
> Furthermore, the reviewer’s note on Line 107 and Figure 2 (pre-trained LLMs generating text actions) also needs clarification. **In our response, we did not claim that our X-ICM framework is strictly learning-based or learning-free. Our work focuses on systematically evaluating the cross-task generalization of existing VLA methods, regardless of whether they are learning-based or learning-free.** The X-ICM framework integrates a learning-based dynamics diffusion module (to improve cross-task sample selection) and a learning-free LLM module (leveraging inherent in-context generalization capabilities). This design is tailored to address cross-task generalization and is evaluated against all existing learning-based or learning-free VLA methods.
>
> &nbsp;
>
> **[Discussion] Issue #2: Addressing Cross-Task Generalization and Training Assumptions.**
>
> (1). Clarification on Cross-Embodiment Transfer:
>
> Regarding the reviewer’s comment in issue #2: “In addressing cross-task generalization, the authors mention that their dynamics diffusion model can be replaced by an off-the-shelf feature extractor like DINOv2 to enable transfer without training,”, **this pertains to cross-embodiment generalization**. We offer this as a trivial solution for transferring X-ICM to new embodiments without training. **More importantly, our work does not claim that X-ICM aims to achieve training-free generalization to arbitrary new embodiments**, as this remains an open VLA challenge outside our scope.
>
> (2). Justification for Training on Seen Tasks:
>
> The reviewer’s comment: “Training on seen tasks weakens the contribution,” warrants clarification. **We clarify that training the dynamics diffusion module on our cross-task benchmark is reasonable**. For our newly developed cross-task benchmark, existing learning-free VLA methods (e.g., VoxPoser) require task/scene-specific adaptations and underperform (Table 2). Learning-based methods achieve better results, but they must be fine-tuned on seen task data (due to the well-known embodiment gap). Training our dynamics module on seen data aligns with these practices and ensures fair comparison.
>
> &nbsp;
>
> We hope these clarifications address your concerns. Our work advances VLA research with a robust **cross-task benchmark** and **framework for cross-task generalization**, offering valuable contributions. We welcome further feedback and are committed to addressing it promptly.
>
>
> Thank you again for your detailed feedback and consideration.
>
>
> Sincerely,
>
> All Authors.

---

> > ### Comment · Reviewer_Qo7p · 2025-08-07
> >
> > Thanks for the clarification, which addresses most of my concerns, for example, the scope of this paper.

---

> > > ### Author Response · Authors · 2025-08-07
> > >
> > > Dear Reviewer Qo7p,
> > >
> > > Thank you for taking your time to read our new responses. We are pleased to hear that our clarifications have addressed most of your concerns, particularly our explanation of the distinction between the cross-embodiment problem and our cross-task problem, which has clarified the generalization this paper aims to achieve.
> > >
> > > If you have any further questions or feedback, please feel free to contact us. We will respond promptly. Thank you very much!
> > >
> > > Sincerely,
> > >
> > > The Authors

---

### Official Review · Reviewer_WCn9 · 2025-07-02

**Clarity:** 3
**Significance:** 3
**Originality:** 1
**Rating:** 5
**Confidence:** 5

**Summary:**

The paper proposes a new benchmark for cross-task generalization in VLAs they call AGNOSTOS. It is built to supplement 18 training tasks from RLBench to provide additional environments for evaluation. The authors then proceed to fine-tune existing VLA models on their training tasks and test for cross-task generalization. They discover that current methods do not do well on these tasks (as expected).

To start overcoming the limitations observed in current VLA models, the paper additionally proposes a mechanism for using off-the-shelf VLMs/LLMs with the RLBench data for in context learning. They dub their method X-ICM. This method leverages the cross-task generalization capabilities of Large Language Models (LLMs) by conditioning them on "in-context demonstrations" from seen tasks to predict action sequences for unseen tasks. The key insight is that this only works if appropriate in-context demonstrations are provided and to obtain such demonstrations they train a goal prediction diffusion model on the dataset (which is afterwards used for retrieval).

**Questions:**

Please address the weaknesses discussed above. In particular questiones regarding:
- Statistical significance of results and providing expanded per-task results.
- Questions regarding in-domain performance to help as a gauge for how well methods perform.
- Questions regarding oracle performance on the proposed tasks.

**Ethical Concerns:**

["NO or VERY MINOR ethics concerns only"]

**Final Justification:**

The issues brought up in my review have largely been addressed and as mentioned in my comment the additional experiments incl. in domain baselines strengthen the paper. I have adjusted my score accordingly.

**Limitations:**

Limitations are discussed adequately in the discussion section (given hte space constraints).

**Paper Formatting Concerns:**

- The paper includes multiple very large images. The first of them is pasted right into the abstract and creates a very odd first page layout.
- The title includes some form of a logo for some reason?

**Quality:**

3

**Strengths And Weaknesses:**

Pros:
- The community is in need for more benchmarks and the proposed benchmark seems to address a whole in current evaluations.
- The tasks seem to be designed thoughtfully and the split into tasks with partially shared semantics (Level 1) and entirely novel (Level 2) seems good.
- They evaluate a meaningful set of pre-trained models from the literature and provide comprehensive evaluation results.
- The explored in-context learning/policy elicitation technique seems simple and reasonable and is of interest to the community.

Weaknesses:
- The reviewer understands that large scale evaluations are costly. Still 25 rollouts with three repetitions for each method seems very little to judge statistical significance of results.
- The real world experiments seem week / are relatively simple pick and place tasks.
- It is unclear how good the in-context learning really is given that all methods score fairly low on most of the tasks (and no per task variances are given?). To calibrate it seems important to also give reference numbers for all methods on in-domain tasks. Where maybe the in-context learning is worse but it would help calibrate how far off the best possible result it is. It would also have been helpful to include some form of oracle score if possible (i.e. how good could a policy get at the tasks given X demonstrations)
- Paper formatting seems off, see also comments below.
- In some sense the benchmark is largely derivative of RLBench, most of it is "just" adding additional tasks to RLBench.
- Details on how open models were fine-tuned seem scarce.
- The retrieval of in-context examples seems relatively involved. It is unclear if such a complicated pipeline is truly necessary. No ablations / comparisons to simpler strategies is given.
- The paper is mainly a systems paper proposing a new benchmark. Without the benchmark it would not pass the acceptance bar.
- Some sentences seem ungrammatical, e.g. just from the first page:
line 28: "This capability-cross-task zero-shot generalization-is essential for real-world deployment, where robots are expected to tackle novel tasks as they arise dynamically." Should probably read something like "The capability of cross-task (zero-shot) generalization is essential for real world deployment, [..]"
line 30: "Despite the rapid progress in VLA research, [...] has focused on generalization testing in [...]" -> "Despite the rapid progress of research into VLAs [...] has focused on testing generalization". VLA is not a noun, "generalization testing" isn't a thing.
line 34: "the generalization of VLA models." -> "the generalization capabilities of VLA models".
line 60: "Guided by the learned dynamics, the dynamics-aware prompts can be formed [...]" -> "Guided by the learned dynamics, dynamics-aware prompts can be formed [...]"

---

> ### Author Rebuttal · Authors · 2025-07-26
>
> **Dear Reviewer:**
>
> We sincerely thank you for recognizing the value of our AGNOSTOS benchmark in addressing a gap in current VLA evaluations, its thoughtful task design, and our comprehensive evaluation of pre-trained models. We believe that our clarifications below can address all your concerns.
>
> &nbsp;
>
> **Q.1 Statistical significance of results.**
>
> To address concerns about the statistical significance of our results, we expanded the evaluation to include 6 different seeds for each task, with 50 rollouts per seed, resulting in a total of 300 rollouts per task. Due to the rebuttal word limit, we present updated results for RVT2 (in-domain VLA), R3M-Align (human-video VLA), OpenVLA, $\pi_0$ (foundation VLA), and our X-ICM (7B) across the 23 unseen tasks, as shown below. Task IDs (T1–T23) correspond to those in Table 2 of our paper.
>
> The updated results confirm that the average success rates and standard deviations for Level-1, Level-2, and all tasks remain consistent with our original findings, with no statistically significant differences. Per-task standard deviations are included to provide transparency. High variance in some tasks is attributed to the RLBench environment’s variable scene configurations, a similar phenomenon also observed in Table I of the RVT2 paper [33].
>
> | - | T1 | T2 | T3 | T4 | T5 | T6 | T7 | T8 | T9 | T10 | T11 | T12 | T13 |
> | :---: | :---: | :---: | :---: | :---: | :---: | :---: | :---: | :---: | :---: | :---: | :---: | :---: | :---: |
> | RVT2 | 0.0 (0.0) | 0.0 (0.0) | 0.0 (0.0) | 20.0 (0.0) | 40.0 (6.9) | 1.3 (2.3) | 65.3 (2.3) | 2.7 (2.3) | 1.3 (2.3) | 0.0 (0.0) | 2.7 (2.3) | 6.7 (2.3) | 34.7 (2.3) |
> | R3M-Align | 0.0 (0.0) | 5.3 (2.3) | 46.7 (4.6) | 26.7 (2.3) | 24.0 (6.9) | 0.0 (0.0) | 53.3 (4.6) | 0.0 (0.0) | 5.3 (2.3) | 0.0 (0.0) | 0.0 (0.0) | 2.7 (2.3) | 0.0 (0.0) |
> | OpenVLA | 0.0 (0.0) |	6.0 (4.6) | 37.8 (6.1) | 41.1 (8.0) | 56.4 (2.3) | 0.0 (0.0) | 52.8 (12.2) | 13.0 (8.0) | 1.3 (2.3) | 1.3 (2.3) | 0.0 (0.0) | 11.3 (4.1) | 0.0 (0.0) |
> | $\pi_0$ | 0.0 (0.0) | 4.8 (1.6) | 84.8 (3.9) | 25.6 (8.6) | 40.0 (3.5) | 1.6 (1.9) | 64.8 (2.9) | 20.0 (4.3) | 7.2 (2.9) | 0.8 (1.6) | 0.0 (0.0) | 32.0 (4.3) | 0.8 (1.6)|
> | **X-ICM (7B)** | 1.6 (1.9) | 25.6 (7.8) | 22.4 (1.9) | 44.0 (4.3) | 34.4 (3.2) | 58.4 (7.8) | 47.2 (12.7) | 57.6 (7.8) | 50.4 (8.8) | 1.6 (1.9) | 0.0 (0.0) | 8.0 (7.1) | 16.8 (5.8) |
>
> | - | T14 | T15 | T16 | T17 | T18 | T19 | T20 | T21 | T22 | T23 | **Level-1 avg (std)** | **Level-2 avg (std)** | **All avg (std)** |
> | :---: | :---: | :---: | :---: | :---: | :---: | :---: | :---: | :---: | :---: | :---: | :---: | :---: | :---: |
> | RVT2 | 26.7 (9.2) | 37.3 (4.6) | 0.0 (0.0) | 0.0 (0.0) | 0.0 (0.0) | 0.0 (0.0) | 0.0 (0.0) | 1.3 (2.3) | 0.0 (0.0) | 2.7 (2.3) | 13.3 (0.6) | 6.6 (1.3) | 10.4 (0.4) |
> | R3M-Align | 89.3 (4.6) | 0.0 (0.0) | 2.7 (2.3) | 0.0 (0.0) | 0.0 (0.0) | 0.0 (0.0) | 1.3 (2.3) | 10.7 (4.6) | 4.0 (0.0) | 0.0 (0.0) | 12.2 (0.7) | 11.0 (0.5) | 11.7 (0.3) |
> | OpenVLA | 75.8 (2.3) | 0.0 (0.0) | 0.0 (0.0) | 0.0 (0.0) | 0.0 (0.0) | 0.0 (0.0) | 8.7 (2.3) | 8.3 (4.1) | 4.1 (2.3) | 0.0 (0.0) | 17.0 (1.1) | 9.7 (0.8) | 13.8 (0.9) |
> | $\pi_0$ | 96.8 (3.9) | 0.0 (0.0) | 1.6 (1.9) | 0.0 (0.0) | 0.0 (0.0) | 0.0 (0.0) | 16.0 (4.4) | 6.4 (4.8) | 1.6 (2.0) | 0.0 (0.0) | 21.7 (0.4) | 12.2 (0.7) | 17.6 (0.4) |
> | **X-ICM (7B)** | 98.4 (1.9) | 17.6 (7.4) | 7.2 (3.9) | 8.8 (1.6) | 0.0 (0.0) | 6.4 (1.9) | 16.8 (2.9) | 2.4 (1.9) | 5.6 (1.9) | 3.20 (2.9) | **28.3 (1.0)** | **16.6 (1.2)** | **23.2 (1.1)** |
>
> &nbsp;
>
> **Q.2 More complex real-world tasks.**
>
> To address concerns about the simplicity of real-world experiments, we evaluated two additional non-pick-and-place tasks: "open the drawer" and "unscrew the bottle cap." To support these tasks, we expanded the real-world seen task set by including "pull out the middle block," which involves a pulling action similar to "open the drawer." Following the real-world testing protocol described in our paper, we achieved success rates of 20% for "open the drawer" and 0% for "unscrew the bottle cap" across 20 rollouts per task.
>
> These results align with expectations: the pulling action in "open the drawer" resembles that of "pull out the middle block," enabling partial success, whereas "unscrew the bottle cap" involves a unique twisting action absent from our seen task set, leading to failure. If we can expand the scale of real-world data collection (ie, seen tasks are diverse enough), the non-pick-and-place tasks could be solved to a certain extent. Our simulation results (i.e., Table 2) demonstrate that increasing the diversity of seen tasks improves performance on non-pick-and-place tasks.
>
> &nbsp;
>
> **Q.3 In-domain and oracle cross-domain performance.**
>
> To address concerns about evaluating the effectiveness of in-context learning, we provide performance metrics for RVT2, R3M-Align, OpenVLA, $\pi_0$, and our X-ICM (72B) on 18 in-domain (seen) tasks, as shown below, alongside their performance on 23 unseen tasks. These results demonstrate that all methods achieve significantly higher success rates on in-domain tasks after fine-tuning on seen task demonstrations, highlighting the challenge of cross-task zero-shot generalization.
>
> For X-ICM, we use 18 within-task demonstrations to construct in-context prompts for in-domain tasks, yielding a 60.4% average success rate, which is substantially higher than its 30.1% on unseen tasks. However, our X-ICM's performance on in-domain tasks is lower than that of learning-based VLA methods like $\pi_0$ (70.5%). This gap is expected, as X-ICM relies solely on the LLM’s in-context learning capability without fine-tuning.
>
> To estimate the oracle performance of X-ICM on the 23 unseen tasks, we conducted a one-shot experiment, providing a single within-task demonstration per unseen task, selected based on semantic similarity using our dynamics-guided sample selection strategy. This achieved an average success rate of 42.8%, which can serve as an upper limit reference for cross-task zero-shot methods (our zero-shot performance: 30.1%).
>
>
> | - | **18 seen/in-domain** avg (std) | **23 unseen** avg (std) |
> | :---: | :---: | :---: |
> | RVT2 | 82.3 (0.7) | 10.3 (0.6) |
> | R3M-Align | 59.2 (0.8) | 12.2 (0.3) |
> | OpenVLA | 63.1 (1.4) | 13.6 (0.8) |
> | $\pi_0$ | 70.5 (0.9) | 17.5 (0.4) |
> | **X-ICM (72B)** | 60.4 (0.7) | **30.1 (1.0)** |
>
> &nbsp;
>
> **Q.4 paper format and grammar correction.**
>
> Thank you for the reminder on the paper format. We will move Figure 1 to the end of the abstract section. And we will remove the logo from the title of the paper. We will correct grammar issues based on your useful suggestions.
>
> &nbsp;
>
> **Q.5 Why did we choose RLBench instead of a new simulation?**
>
> We clarify that our contribution extends significantly beyond adding tasks to RLBench.
>
> Our primary contribution lies in introducing a novel evaluation framework focused on cross-task, zero-shot generalization. This framework, supported by a large-scale evaluation of different types of VLA models, provides the first comprehensive analysis of zero-shot generalization capabilities. Our findings (e.g., Table 2) reveal critical limitations in current VLA models and provide the community with a standardized tool to measure future progress.
>
> In addition, our choice of RLBench as the foundation for our benchmark was deliberate. RLBench is one of the most widely used manipulation environments in the community, and it provides a rich ecosystem where many influential methods are already validated. Building a benchmark on this environment allows for immediate, authoritative comparisons against prior work.
>
> &nbsp;
>
> **Q.6 Finetuning Details.**
>
> Thank you for pointing out the need for more detailed descriptions of the fine-tuning process for existing VLA methods. For each VLA method, we follow the official fine-tuning guidelines and explain the details if we use different setups. We will elaborate on all the fine-tuning details in the revised version. Thank you so much for this suggestion.
>
> &nbsp;
>
> **Q.7 More in-context sample selection baselines.**
>
> We compared our dynamics diffusion model with two simpler baselines: CLIP feature-based and DINOv2 feature-based sample selection, alongside a random selection strategy. The table below reports the average success rates and standard deviations for 23 unseen tasks.
> Both CLIP and DINOv2 baselines outperform random selection in terms of average performance and stability, but they fall short of our dynamics diffusion model. This is because CLIP and DINOv2 can only capture static visual features (only a single initial frame is available during testing each unseen task), which fail to capture the temporal dynamics critical for cross-task generalization. In contrast, our dynamics diffusion model leverages future visual predictions to learn temporal dynamics, which is more effective in sample selection, as detailed in Section 4.2.
>
> | - | **Level-1 avg (std)** | **Level-2 avg (std)** | **All avg (std)** |
> | :---: | :---: | :---: | :---: |
> | **Random** | 30.7 (4.7) | 18.0 (2.2) | 25.2 (3.2) |
> | **CLIP feature** | 32.1 (3.5) | 18.9 (2.6) | 26.4 (2.9) |
> | **DINOv2 feature** | 33.7 (3.1) | 19.3 (2.0) | 27.4 (2.5) |
> | **Dynamics Diffusion (Ours)** | **37.6 (1.4)** | **20.3 (1.7)** | **30.1 (1.0)** |

---

> > ### Comment · Reviewer_WCn9 · 2025-08-07
> >
> > I want to thank the authors for the substantial effort they put in to provide additional experiments and baselines in the rebuttal period.
> >
> > As mentioned in my initial review I don think the benchmark is useful and the paper provides some insights that may prove useful for future studies in this growing field.
> >
> > The additional experiments that also showcase stronger baselines/in-domain performance do paint a much more complete picture and will be great to include in the final version.
> >
> > Most of my concerns have been addressed and so I would recommend acceptance.

---

> > > ### Author Response · Authors · 2025-08-07
> > >
> > > Dear Reviewer WCn9,
> > >
> > > Thank you for the time and effort in reviewing our work. We are delighted to hear that your concerns have been addressed. We will incorporate the additional experimental results and baselines into our revision.
> > >
> > > We greatly appreciate your recognition of the value of our benchmark and your indication that our paper will inspire more future work in the VLA field. We are very encouraged by your recognition and the recommendation for acceptance.
> > >
> > > Sincerely,
> > >
> > > The Authors

---

### Note · Authors · 2025-08-13

Dear AC and Reviewers,

We sincerely appreciate the reviewers' engagement in a constructive discussion and their recognition of our work. Below, we summarize key feedback for the AC and reviewers.

- Reviewer WCn9 acknowledged the necessity and effectiveness of our proposed Cross-task Zero-shot Benchmark for the manipulation community, praising its thoughtful task design, comprehensive model evaluation, and novel, reasonable model design. Post-rebuttal, the reviewer confirmed their concerns were addressed and recommends acceptance.

- Reviewer Qo7p recognized the well-motivated contribution of our benchmark as a platform for future research, its effective methodology, and comprehensive simulation and real-world experiments. Our initial rebuttal addressed most concerns, particularly regarding dynamics-guided sample selection and the zero-shot definition. The reviewer raised their score and indicated openness to a higher rating if remaining concerns about generalization were clarified. Our subsequent response clarified these concerns, and we are delighted that the reviewer acknowledged our new clarifications. We respectfully hope that the reviewer will consider our paper for acceptance.

- Reviewer Xb1Z noted that our benchmark is a timely and valuable contribution, with a clear, well-written paper, a thorough literature review, and significant exploration of cross-task in-context learning in robotics. After the rebuttal, the reviewer confirmed their questions were fully addressed and recommended acceptance.

- Reviewer KVSd valued our benchmark’s evaluation of cross-task zero-shot generalization in manipulation and noted the impressive experimental results compared to existing state-of-the-art methods. Post-rebuttal, the reviewer confirmed most concerns were addressed, raising their rating to acceptance.

We thank the AC and reviewers for their thorough feedback and efforts. We are committed to incorporating all clarifications, discussions, and comparisons into our revision based on the reviewers’ suggestions.

Sincerely,

Authors of Submission 398

---

### Decision · Program_Chairs · 2025-09-17

**Decision:**

Accept (poster)

**Comment:**

The paper presents a benchmark for cross-task zero-shot manipulation built from RLBench (18 seen training tasks; 23 unseen evaluation tasks split into Level-1 with partial semantic overlap and Level-2 with novel compositions). Baselines show poor transfer to unseen tasks. The paper then introduces an approach that elicits actions from a pretrained LLM using in-context demonstrations retrieved from seen tasks by a dynamics-guided diffusion model that predicts goal images from a single initial frame to capture temporal dynamics and outperforms the baselines. The rebuttal strengthened the evidence: reporting in-domain vs unseen performance and a one-shot “oracle”, comparing retrieval against CLIP/DINOv2/random, and adding small real-world and long-horizon studies.

The work's strengths are a timely benchmark that standardises evaluation for cross-task generalisation, a thoughtful task split, and broad empirical coverage that was materially improved during rebuttal, including clear ablations demonstrating the value of dynamics-guided retrieval. Its weaknesses are that the benchmark merely extends RLBench and the methodological advance over prior in-context pipelines (e.g., RoboPrompt) is modest and subtle. The pipeline is also comparatively complex and relies on training the diffusion component on seen tasks, which raises portability questions. Furthermore, elements of the action decoding and formatting required clarification and real-world and long-horizon performance remains preliminary.

Even so, the benchmark fills a genuine gap and the strengthened experiments address the principal concerns. I therefore recommend accepting, with the camera-ready expected to integrate the ablations, the expanded real-world and long-horizon results and additional experiments and clarifications provided in the rebuttal.